# Efficiency optimization for large-scale droplet-based electricity generator arrays with integrated microsupercapacitor arrays

Zheng Li ⬤ , Shiqian Chen, Yujie Fu & Jiantong Li ⬤ ✉

Droplet-based electricity generators are lightweight and nearly metal-free, making them promising for hydraulic power applications. However, two critical challenges hinder their practical application: significant performance degradation, potentially up to 90%, in existing small-scale integrated panels, and low efficiency, often less than 2%, in storing the irregular high-voltage pulsed electricity produced by large-scale arrays. Here, we demonstrate that by tailoring the bottom electrodes so that their area is comparable to the spread area of the impinging water droplets, we double the average output power of individual cells and fabricate large-scale (30-cell) arrays that achieve approximately 2.5 times higher power than state-of-the-art arrays. Furthermore, without using any power management chip, we integrate a large-scale (400-cell) micro-supercapacitor array to store the irregular high-voltage electricity produced by the 30-cell generator array at an efficiency of 21.8%. The integration of large-scale electricity generator arrays and micro-supercapacitor arrays forms a simple, chipless, self-charging power system with an output power of 81.2 µW, which is 27 times higher than current systems based on 30-cell arrays. This work provides important insights towards practical applications of droplet-based electricity generators.

As a renewable resource with vast availability on Earth, water energy harnesses the mechanical movement of natural water, such as raindrops, river flows, and ocean waves, to generate sustainable power[1]. Recently, a variety of emerging technologies have been developed to harvest water energy, including electromagnetic harvesters[2,3], electroactive polymer harvesters[4–6], moisture-enabled electric generators[7–10], and liquid-solid triboelectric nanogenerators (TENGs)[11–15]. Among the liquid-solid TENGs, droplet-based electricity generators (DEGs)[12] have garnered significant attention because of their simple structure, low cost and high power density[12]. A DEG utilizes the falling water droplets to impinge a triboelectric polymer, and the spread water droplets can effectively collect triboelectric charges on the polymer surfaces to produce instantaneous (pulsed) electricity at a high peak power density of ~50 W m$^{-2}$ and a high average power density of ~50 mW m$^{-2}$. Almost fully made of polymeric (metal-free,

corrosion-resistant) materials, the DEGs have great potential to replace conventional electromagnetic generators in emerging fields of hydraulic power, such as ocean wave power where the harsh marine environment causes severe corrosion to the metal electromagnetic generators and significantly increase the maintenance cost[16]. Since their invention[12] in 2020, various methods have been developed aiming to improve the output performance of DEGs at the cell level, including device structures such as co-planar electrodes and single-electrode devices[17–19], injecting additional surface charge[20], tuning droplet dynamics[21–24] and modulating surface roughness[25,26]. However, up to date most advance is merely achieved for modulating the output pulse form of individual DEGs. For example, the output peak voltage has increased from the general 150 V to over 1000 V[23,24], the pulse period can be adjusted in a wide range from milliseconds down to microseconds[23,27], and the direct output can be changed from

Division of Electronics and Embedded Systems, School of Electrical Engineering and Computer Science, KTH Royal Institute of Technology, Stockholm, Sweden. ✉e-mail: jiantong@kth.se

alternating current (AC) to direct current (DC)[28]. On the contrary, nearly none of the present DEGs can attain an effective (average) power density, the most important performance indicator for energy harvesting, surpassing the original value of 50 mW m$^{-2}$ of the first DEGs[12]. Moreover, like the evolution from solar cells to solar panels, the most important step for practical applications of DEGs is to scale up individual cells into large-scale DEG panels and arrays to obtain sufficient overall output power[29,30]. However, the latest research[29,30] has indicated several critical challenges in performance scaling. First, due to the AC output of the general DEGs, in principle every DEG cell needs to be equipped with a full-wave rectifier to transfer the AC output to DC to avoid the destructive interference with other DEGs[31]. This, however, significantly increases the complexity and cost for the fabrication and maintenance. To make a trade off, Ye et al.[30] propose a strategy to mitigate the intercell interference through reducing the dripping frequency to 3 Hz, which allows three DEGs to share one rectifier. As a result, a 30-cell DEG array (consisting of 10 panels and each panel comprising 3 cells) has been fabricated with the use of only 10 rectifiers to achieve an overall effective output power of 152 μW. Second, different from the AC output-induced intercell interference, the large panel-level parasitic capacitance also severely degrades the output performance of DEG panels. Xu et al.[29] have fabricated a 9-cell DEG panel with the use of only one single rectifier. All the DEG cells share the common device structure of a coplanar electrode design where both the two electrodes are located on the top surface of the substrate without any overlapping, aiming to reduce the parasitic capacitance and maximize the pack density of the panels with mini-mized dead area. However, the peak power density in the panel can only achieve 22.5 W m$^{-2}$, just 27% of that of single cells (83.0 W m$^{-2}$). In other words, the 9-cell DEG panel even has inferior performance to one single DEG cell. After every DEG cell is equipped with a full-bridge rectifier, the effective output power of the multiple-rectifier panel even decreases by about 30%, as compared with the single-rectifier panel[29]. It suggests that the severe degradation of output power density in the panel should not be ascribed to the AC output-induced intercell interference, but to the large panel-level parasitic capacitance as dis-cussed in the following section. Last and most importantly, it is very challenging to effectively store the output pulsed electricity of large-scale DEG panels. DEGs, as well as almost all the other advanced TENGs developed recently, produce high-voltage instantaneous pulsed elec-tricity typically with peak voltage >100 V and pulse period at the level of 10 ms. Conventional energy storage devices, such as batteries and capacitors, can only store such electricity at very low efficiency of <2%[32,33]. To address the issue, various power management strategies have been developed, including transformers, switch capacitors, and buck converters, but their energy storage efficiency is still <6%[33]. Very recently, Wu et al.[33] have demonstrated an energy storage efficiency of ~5% with a power management circuit comprising capacitor, inductor, diode and mainly a needle-based discharge switch, where the gap between a pair of needles is tailored so that once TENGs reach their maximum output voltage, the switch closes, i.e., the needle pair breaks down synchronously to achieve the maximum energy release from the TENGs[34]. Through further tailoring the TENG device structure to lower their output voltage and improving the atmosphere at the needle tip gap to reduce energy loss during discharge, it is even possible to increase the energy storage efficiency to 42.5% for regular mechanical stimuli[33]. When it comes to irregular mechanical stimuli, the switch circuit needs to incorporate with a commercial power management chip, together with well-designed auxiliary circuits, to retain the high energy storage efficiency of 39.8%[33]. However, due to the need for a series of optimizations of working conditions for the TENGs, needle-based discharge switch, and case-dependent circuit design, such an energy storage strategy can hardly apply to large-scale DEG panels or arrays. The intrinsic randomness and variation of the droplet size,

impinging velocity and frequency, and landing position (with respect to electrodes) significantly impact the output stability of individual DEGs[21,35]. When multiple DEGs work simultaneously in a large-scale array, the constructive and destructive interference significantly increases the irregularity and unpredictability of the overall outputs, making it almost impossible to optimize the component and circuit design in the present power management strategies to obtain high energy storage efficiency. As a matter of fact, in the present research, a 30-cell DEG array can attain an overall average power as high as 152 μW, but it can only charge a capacitor at a low power of about 3.0 μW[30], corresponding to energy storage efficiency of only 2.0%. The low energy storage efficiency, together with the unscaled output power, generates critical challenges in using large-scale DEG arrays for energy harvesting from natural water.

Here, we show a generic strategy for high-efficiency energy harvesting based on large-scale DEG arrays (Fig. 1a). First, we unveil the currently ignored critical factor for the panel-level parasitic capacitance that may account for severe degradation in output power of DEGs integrated in a panel[29]. In spite of a lot of efforts in the literature to reduce the parasitic capacitance through reducing the footprint area of top electrodes[30] or the overlapping area between top and bottom electrodes[29], the impact of bottom electrodes is ignored so far. In this work we have found that the bottom electrode has dual functions. A sufficiently large area of bottom electrode is necessary to maximize the bulk capacitance between the spread water droplet and bottom electrode to ensure high output power of the DEGs. However, too large a bottom electrode area can evidently increase the parasitic capacitance in the circuits to decrease the output power. As a result, the DEG output is maximized when the bottom electrode area is comparable to the spread area of the impinging droplets. Based on this finding, simply through adapting the global bottom electrodes (GBEs) to localized bottom electrodes (LBEs) with optimized electrode area, the output average power density of individual DEGs can increase by almost 4 times, from 28.3 mW m$^{-2}$ to 109.0 mW m$^{-2}$. It is 2.2 times higher than that (49.4 mW m$^{-2}$) of the original DEGs[12]. More importantly, the LBE design enables us to integrate up to 5 DEGs into one panel with only one full-bridge rectifier to attain an overall average power of 85.9 μW, 2.6 times higher than our own single DEG cells (32.7 μW) or over 6 times the original DEG cells (13.4 μW)[12]. When 6 panels are integrated into a 30-cell DEG array, the overall average power reaches 371.8 μW, more than twice that (152 μW) of the previous 30-cell DEG arrays[30]. Furthermore, as expected, the large-scale DEG array gives rise to very irregular electricity output (Fig. 1b). In order to increase the energy storage efficiency (ESE), large-scale ultrafast metal-free micro-supercapacitor (MSC) arrays with up to 400 cells are fabricated simply through the combination between direct ink writing and laser scribing. The MSC arrays have an expected working voltage window up to 640 V and a high charge rate up to 2000 V s$^{-1}$ (Fig. 1c). In no need of any extra electronic component or circuit design, they can store the irregular output electricity of DEG arrays at the power of 81.2 μW, more than 27 times higher than the store power in the existing best DEG arrays in the literature[30]. In other words, 21.8% of the maximum output power (371 μW) of the DEG arrays has been effectively stored (Fig. 1d). The ESE is ~11 times higher than that (2.0%) of direct use of commercial capacitors in the existing DEG arrays[30], and comparable to that (39.8%) of the integration of commercial chips with case-dependent circuit design in TENGs[33]. After the 30-cell DEG array charges the 400-cell MSC array for only 30 s, the inte-grated self-charging power system (SCPS) can supply a light emitting diode (LED) to work continuously for 60 s, suggesting the promise of the strategy to integrate large-scale DEG arrays with large-scale ultrafast MSC arrays to build SCPSs for high-efficiency energy har-vesting from natural water towards practical applications.

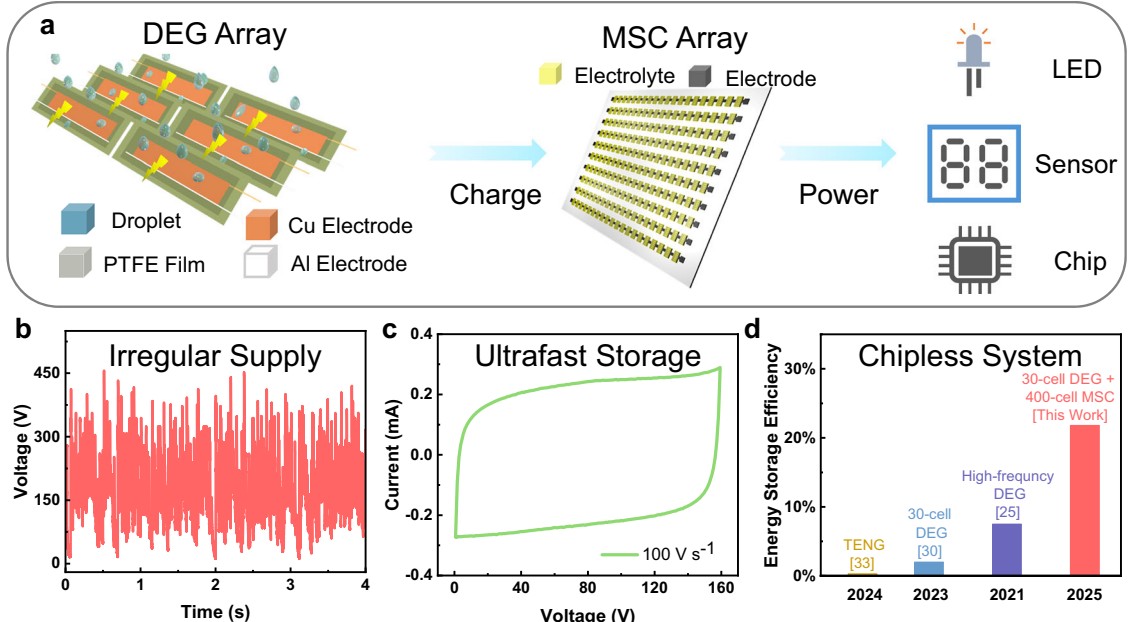

**Fig. 1 | Key concepts of our self-charging power system. a** Schematic of our self-charging power systems (SCPS) featuring integration of large-scale droplet-based electricity generator (DEG) arrays with large-scale microsupercapacitor (MSC) arrays (PTFE: polytetrafluoroethylene, LED: light emitting diode). **b** Irregular high-voltage of a 30-cell DEG panel array. **c** Fast charge-discharge performance (cyclic voltammetry (CV) curve at 100 V s⁻¹) of a 100-cell MSC sub-array. **d** Comparison of the energy storage efficiency between our system and others[25,30,33], including those using triboelectric nanogenerators (TENGs). Source data are provided as a Source Data file.

## Results

### Optimization of droplet-based electricity generator cells with localized bottom electrodes

In general, a DEG comprises a triboelectric polymer (such as poly-tetrafluoroethylene, PTFE) film equipped with a wire-shaped top electrode and a GBE[12]. Our LBE-DEGs have similar device structure to GBE-DEGs but with localized square bottom electrodes (Figs. 2a, S1a). They also exhibit similar voltage-time characteristic curves (AC signals) as GBE-DEGs (Fig. S1b). Prior to droplet impact, the negatively charged PTFE induces a positive charge on the bottom electrode. At a certain time ($t_1$ in Fig. S1), the impinging droplet spreads to its maximum area and contacts the top electrode to quickly transfer the positive charge from the bottom electrode to the top electrode and generate a high instantaneous voltage. During the retraction of the droplet, the positive charge gradually returns to the bottom electrode, resulting in a transition of the DEG output from positive to negative voltage. Finally, the droplet detaches from the top electrodes (at the time of $t_3$ in Fig. S1), and both the charge and output voltage drop to zero[12]. However, under the same testing conditions, our LBE-DEGs exhibit almost twice higher output peak voltage than the GBE-DEGs, no matter whether there is overlapping between bottom and top electrodes (Fig. 2b). A systematic study (Fig. 2c) indicates that even with the minimized top electrodes (5 cm × 0.2 cm), the peak voltage of LBE-DEGs still strongly varies with bottom electrode area and reaches the maximum when the bottom electrode area $S_{BE}$ is close to the maximum spread area $S_{D,max}$ of the impinging droplets (around 3 cm² in this work). Moreover, we also investigate the LBE effects under altered testing conditions and liquid properties, such as environmental temperature and humidity (Fig. S2), droplet falling height (Fig. S3), impact angle (Fig. S4), droplet volume (Fig. S5), and water type or composition (Fig. S6). In general, the DEG performance varies with the testing conditions and water properties. For example, the increase in ion concentration of droplet water degrades the DEG output performance (Fig. S6), which is generally consistent with previous studies[23,29]. However, it is important to note that the LBE effects remain in all the cases (Fig. S2–6), and the optimal performance always takes place at $S_{BE} \approx S_{D,max}$. Because the LBE-DEGs share the same top electrode structure and droplet dynamics as the GBE-DEGs, the significantly increased output voltage should be ascribed to the optimized bottom electrodes, as explained below.

As shown in the simplified circuit model[12,21,23] in Fig. 2a, a DEG mainly comprises three types of capacitors: bulk capacitor $C_B$, device-induced parasitic capacitor $C_{P,D}$ and circuit-induced parasitic capacitor $C_{P,C}$. When a falling droplet impacts the PTFE surface without contacting the top electrode, the bulk capacitor $C_B$ is formed between the spread water droplet (as the top electrode) and the copper bottom electrode, with the PTFE film as the dielectric. During the droplet impinging and spreading, mechanical energy is converted into electrical energy and stored in $C_B$. Once the droplet contacts the top electrode, the energy stored in $C_B$ is released to generate a high instantaneous output voltage $V_{peak}$. However, due to the existence of parasitic capacitance $C_{P,D}$ (parasitic capacitance between the non-overlapping top and bottom electrodes) and $C_{P,C}$ (parasitic capacitance due to the electrical measurement circuit), a part of the energy stored in $C_B$ is actually transferred to $C_{P,D}$ and $C_{P,C}$, with a relatively lower $V_{peak}$ as (Supplementary Note 1)

$$V_{peak} = \frac{Q}{C_B + C_{P,D} + C_{P,C}} = \frac{U_0 C_B}{C_B + C_{P,D} + C_{P,C}} \quad (1)$$

where $Q = \sigma_S S_{eff}$ is the charge stored in $C_B$ with $\sigma_S$ being the permanent surface charge density on the PTFE film and $S_{eff} = \min(S_{BE}, S_{D,max})$ being the minimal value between the bottom electrode area and the maximum droplet spread area, and $U_0 = \frac{Q}{C_B} = \frac{\sigma_S}{c_d}$ is the intrinsic initial voltage ($c_d = \varepsilon_0 \varepsilon_r / d$ is capacitance per unit area between the droplet and LBE, with $\varepsilon_0 \varepsilon_r$ and $d$ being the dielectric permittivity and thickness of the PTFE film, respectively)[21]. $U_0$ is determined by the dielectric PTFE films and independent of the DEG device structure. In this work, all the DEGs have identical maximum droplet spread area of $S_{D,max} \approx 3$ cm² (Fig. S1a). Therefore, from Eq. (1), the maximization of $V_{peak}$ requires to maximize $C_B$ and meanwhile suppress $C_{P,D}$ and $C_{P,C}$. Based on both direct experimental measurement and finite element method (FEM)

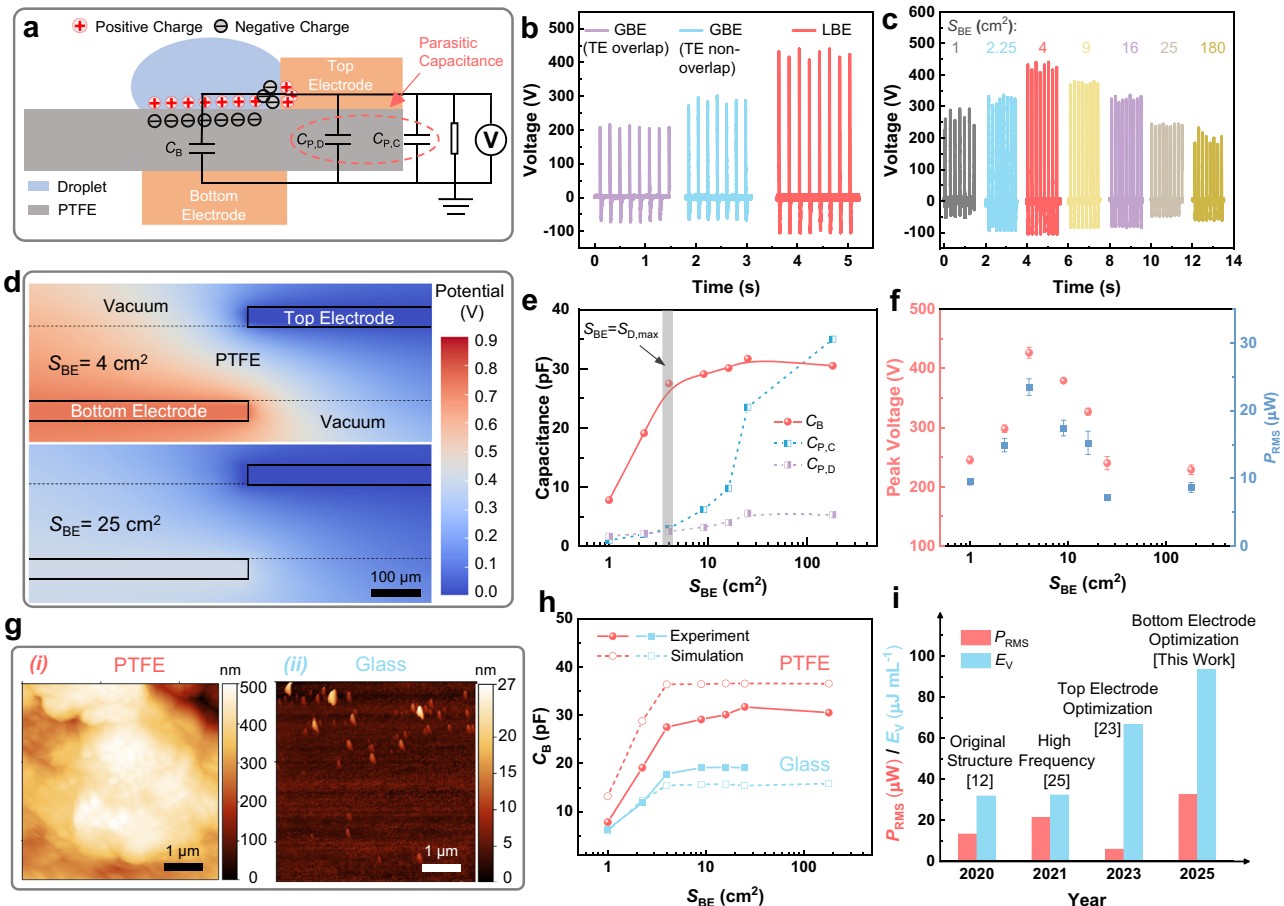

**Fig. 2 | Optimization of DEG cells with localized bottom electrodes. a** Schematic illustration and simplified circuit model of the localized bottom electrode (LBE) design for a DEG cell. The model mainly consists of bulk capacitor $C_B$, device-induced parasitic capacitor $C_{P,D}$, and circuit-induced parasitic capacitor $C_{P,C}$. **b** Output voltage of the DEG cells with different electrode designs: (from left to right) global bottom electrode (GBE) overlapping top electrode (TE), GBE not overlapping TE, and LBE. **c** Output voltage of the LBE-DEG cells with different areas of the bottom electrode $S_{BE}$ (varying from 1 cm × 1 cm to 15 cm × 12 cm). **d** Finite element method (FEM) simulated potential distribution on cross-section of the DEG cells with different $S_{BE}$. **e** Dependence of the bulk capacitance and parasitic capacitance on $S_{BE}$. Within the gray shaded zone, $S_{BE}$ is equal to the maximum droplet spread area ($S_{D,max}$) and the difference between the bulk capacitance and the parasitic capacitance is maximal. **f** Peak voltage $V_{peak}$ and average output power $P_{RMS}$ of the DEG cells with different $S_{BE}$. Error bars represent the standard sample deviation. **g** Atomic force microscopy (AFM) images indicating surface roughness of (i) a PTFE film and (ii) a glass slide. **h** Experimental and simulated dependence of $C_B$ on $S_{BE}$ for PTFE and glass. **i** Average output power $P_{RMS}$ and energy harvesting efficiency $E_V$ of DEG cells developed in recent years[12,23,25]. Source data are provided as a Source Data file.

simulations (Figs. 2d, e, S7a, b), we have found that at the minimized top electrode area (5 cm × 0.2 cm), both $C_B$ and $C_{P,D}$ increase first with the bottom electrode area $S_{BE}$ and then become to saturate (Fig. 2e). As indicated by the FEM simulation results (Fig. 2d), in spite of the lack of overlapping between the top and bottom electrodes in our LBE-DEGs, the fringing effects[36] still induce considerable parasitic capacitance $C_{P,D}$ between the two electrodes. Nevertheless, $C_{P,D}$ saturates much slower than $C_B$, so that when $S_{BE} = S_{D,max}$, $C_B$ has already approached saturation while $C_{P,D}$ stays at a low value. This asynchronous saturation between $C_B$ and $C_{P,D}$ offers opportunities to increase $V_{peak}$ through optimizing the bottom electrode area. Moreover, through adding a series of external parallel capacitors to the LBE-DEG circuit and exploring their effects on $V_{peak}$ (Fig. S8), we have extracted experimentally[23] the circuit-induced parasitic capacitance $C_{P,C}$. As shown in Figs. 2e, S7b, $C_{P,C}$ has a similar behavior as $C_{P,D}$, with the maximum value of $C_B/C_{P,C}$ occurring at $S_{BE} \approx S_{D,max}$. This confirms that $S_{BE} = S_{D,max}$ is the most favored bottom electrode area to maximize $V_{peak}$, where $C_B$ is almost maximized and meanwhile $C_{P,D}$ and $C_{P,C}$ nearly minimized. However, different from $C_B$ and $C_{P,D}$ that saturate at $S_{BE} \leq 25$ cm², there is no clear tendency for $C_{P,C}$ to saturate even at a large $S_{BE}$ of 180 cm² (Fig. 2e), which should account for the

continuous drop of $V_{peak}$ with increasing $S_{BE}$ (Fig. 2c, f). This behavior needs to be carefully considered in future circuit design for water energy harvesting through large-scale DEG arrays where bottom electrodes are significantly increased. In this work, merely through optimizing the bottom electrode area to $S_{BE} = S_{D,max}$ and without any extra circuit design, we have been able to increase the average output power from 8.5 μW for GBE-DEGs (comparable to the value of 13.4 μW of the original DEGs[12]) to 26.6 μW for LBE-DEGs (Fig. 2f). After optimizing the resistance of the load resistors, the maximum output power further increases to 32.7 μW (Fig. S9b). In addition, the deionized (DI) water-driven LBE-DEGs exhibit good stability, retaining 72% of its initial output power after 8 h of continuous operation (Fig. S10). The degradation should be mainly attributed to the accumulation of moisture and droplet residues on the PTFE surface which reduces the interaction area between the sequent droplets and the PTFE surface. The residue issue is more severe when tap water is used instead of DI water. Nevertheless, the DEG performance can be almost fully recovered simply by using paper tissue to clean the PTFE surface (Fig. S10). This implies that the performance degradation is not any intrinsic failure of the DEG devices, but just the alteration of the operating conditions due to long-time interaction between the PTFE

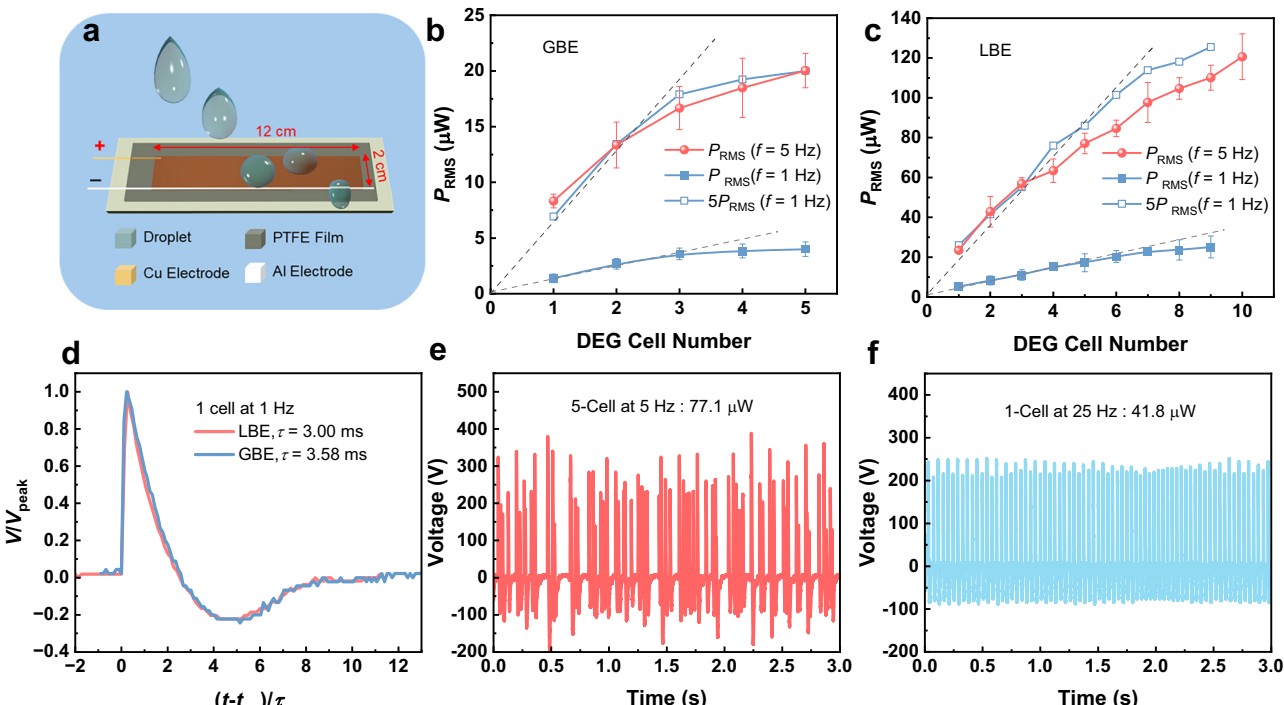

**Fig. 3 | Design and output performance of DEG panels. a** Schematic illustration of an optimized LBE-DEG panel for 5 cells. Average output power of (**b**) GBE and **c** LBE DEG panels against working cell number. Error bars represent the standard sample deviation. **d** Normalized output voltage $V/V_{peak}$ against normalized time $(t-t_{on})/\tau$ for single DEG cells working on GBE and LBE panels, where $t_{on}$ is the time when the pulse starts to occur and $\tau$ is the electrical relaxation time. **e** Output voltage of an optimized 5-cell LBE-DEG panel at frequency $f$ = 5 Hz. **f** Output voltage of an optimized single LBE-DEG cell at $f$ = 25 Hz. Source data are provided as a Source Data file.

surfaces and impinging droplets. So, we believe with proper maintenance the DEGs have great potential to work for weeks or even longer.

In the previous report, Zhou et al.[26] have demonstrated that large surface roughness of the dielectric films can induce more surface charge to improve the DEG performance. Here we would point out the large surface roughness of PTFE films (Fig. 2g) may adversely affect the DEG performance. As shown in Fig. 2h, for PTFE, the experimentally measured $C_B$ is ~20% lower than the simulation value, while the experimental $C_{P,D}$ is nearly twice larger than the simulation results (Fig. S7c). According to Eq. (1), the experimentally decreased $C_B$ and increased $C_{P,D}$ both cause degradation of $V_{peak}$, resulting in an extra voltage loss by ~8% (assuming $C_{P,C}$ does not change). The evident capacitance difference between simulations and experiments might be ascribed to the large surface roughness (the root-mean-square (RMS) roughness is around 88 nm, Fig. 2g(i)) of the PTFE films which is not considered in the FEM models. According to previous research[37], the large surface roughness may either increase or decrease the capacitance under different conditions. In contrast, when the rough PTFE film is replaced with a smooth glass slide with RMS roughness as low as 1.7 nm (Fig. 2g(ii)), both of the simulated $C_B$ and $C_{P,D}$ agree much better with their experimental values (Figs. 2h and S7c). These imply that it is important to comprehensively consider the effects of dielectric surface roughness on the DEG performance.

In short, the simple optimization of bottom electrodes can effectively suppress the adverse parasitic capacitance $C_{P,D}$ and $C_{P,C}$ in the DEGs without evidently reducing the favorable body capacitance $C_B$, so as to significantly improve the output power of the DEGs with similar droplet dynamics and device architecture (Fig. 2i, Table S1). More importantly, the LBE-induced improvement applies to all test conditions (e.g., different droplet size, water type, impinging height, angle, environmental temperature and humidity), as shown in Figs. S2–6. As compared with the previous advanced GBE-DEGs (Fig. 2i), our LBE-DEGs exhibit at least 50% higher average output power (in

comparison with the high-frequency GBE-DEGs[25]), and 40% higher energy harvesting efficiency $E_V$, defined as harvested energy per unit droplet volume (in comparison with the GBE-DEGs with optimized top electrodes)[23]. This knowledge is crucially important for developing large-scale DEG panels which suffer from more severe parasitic capacitance induced by significantly increased bottom electrode area.

## Upscaling to droplet-based electricity generator panels
A single-rectifier DEG panel (Fig. 3a) comprises multiple DEG cells integrated on the same panel (substrate) and sharing the same top electrode, bottom electrode and rectifier. It is a desired device structure for large-scale manufacturing due to their similar fabrication complexity and production cost to a single DEG cell. However, integrating individual DEG cells into a single-rectifier DEG panel often incurs significant energy loss. So far, such energy loss is ascribed only to the AC-induced inter-cell destructive electrical interference because all of the DEG cells only share one common rectifier. As shown in Fig. S11, the output voltage of a 10-cell DEG panel involves strong constructive and destructive electrical interference when all the DEGs work at 5 Hz, although all the droplets were separated from one another throughout the falling and spreading processes. The constructive interference generates randomly many peaks with significantly increased output voltage. They do not really increase the overall output power but increase the difficulty in storing such irregular pulsed electricity, which will be addressed in the following section. Meanwhile, the destructive interference may significantly diminish the output voltage to induce evident energy loss. One potential solution is to equip every individual cell with a rectifier, but this would significantly increase both manufacturing costs and device complexity, as well as causes extra rectifier-induced energy loss[29]. In the literature[30], reducing the working frequency of the DEGs has been supposed to be another effective solution to diminish the inter-cell interference and prevent energy loss. Indeed, as shown in Figs. 3b and S12, S13, when the DEG cells work at frequency as low as

$f = 1$ Hz on a 5-cell panel with GBEs ($15 \times 12$ cm$^2$), the overall average output power increases in excellent proportion to the cell number $n$ when $n \leq 3$, from 1.4 μW for 1 cell to 3.6 μW for 3 cells, suggesting negligible inter-cell interference. When the frequency increases to $f = 5$ Hz, the overall average power also increases almost proportionally with $n$ when $n \leq 3$. In addition, the average power at $f = 5$ Hz is nearly 5 times higher than that at $f = 1$ Hz (Fig. 3b), indicating good scalability of the DEG panel power at working frequency $f < 5$ Hz. However, one should note that at $f = 5$ Hz, the overall average power of the 5-cell panel with global bottom electrodes is only around 20.0 μW (Fig. 3b), which is even lower than that (26.4 μW) of an individual LBE-DEG cell with optimal area $S_{BE} = 3$ cm$^2$ (Fig. 2f). In this work, we have found that like the DEG cells, LBEs also play more important roles in the DEG panels. That is to say, the bottom electrode area of a panel $S_{Panel}$ should be kept as nearly the product of the cell number $n$ and the optimal bottom electrode area of an individual DEG cell $S_{Cell}$, i.e., $S_{Panel} = nS_{Cell}$. As shown in Fig. 3c and Figs. S14, 15, at low frequency $f = 1$ Hz, the overall average power of the LBE panels ($2$ cm $\times 12$ cm) increases proportionally with cell number for $n \leq 6$. At high frequency $f = 5$ Hz, the power still increases with $n$ in approximate proportion for $n \leq 5$. The power reaches 77 μW at $n = 5$ (Fig. 3c), almost 4 times higher than that of GBE-DEG panel (Fig. 3b) at the same conditions ($n = 5$, $f = 5$ Hz). The power at $f = 5$ Hz is also roughly 5 times higher than that at $f = 1$ Hz. The upper limit of $n$ for the proportional power increase has been extended from $n = 3$ for the GBE panels to $n = 5$–6 for the LBE panels. This extension is of significance for the fabrication of single-rectifier DEG panels because more DEG cells can be integrated into the panels to increase the overall output power without significant energy loss. The extension should be ascribed to the shorter period (~33 ms, Fig. S14c) of the output pulses in LBE panels than that in GBE panels (~41 ms, Fig. S12c). A short pulse period allows to accommodate more DEGs in the same panel with diminished electrical interference. The short pulse period in LBE panels also benefits from their low parasitic capacitance $C_P$. According to the scaling law[21], the electrical response curves of DEGs, such as voltage-time ($V$–$t$) curves, collapse when $V$ and $t$ are normalized to $V/V_{peak}$ and $(t - t_{on})/\tau$, respectively, where $t_{on}$ is the time when the pulse starts to occur and $\tau = R(C_B + C_P)$ is electrical relaxation time. In this study, after normalization the voltage response curves for LBE and GBE panels also collapse (Fig. 3d) when $\tau = 3.00$ ms is used for the LBE panel and $\tau = 3.58$ ms for the GBE panel. The values of $\tau$ are in qualitative agreement with the experimental parameters, including $R = 100$ MΩ for the load resistance from the oscilloscope probe impedance, and the experimentally measured $C_B = 27.5$ pF for the LBE panel and $C_B = 30.5$ pF for the GBE panel. It is challenging to experimentally measure $C_P$ for the panels, but one can expect that the LBE panel has smaller $C_P$ than GBE. Because LBE and GBE panels have comparable $C_B$, the ~20% shorter $\tau$ of the LBE panel should be mainly ascribed to its smaller $C_P$. This then accounts for the 20% shorter pulse period of the LBE panel (Fig. S14c), which increases its capability of accommodating more DEGs. As a result, not only do the LBEs attain higher average power for each DEG cell, but they also can accommodate more DEGs in the same panel to further increase the overall output power. Interestingly, we have also found that electrically connecting several small-scale LBE panels in parallel through wires gives rise to also the same output as a large-scale LBE panel of the same total cell number (Fig. S16). For example, connecting a 3-cell panel with 1-cell panel (or a single DEG cell) produces a power of 61.8 μW while a 4-cell panel gives 65.0 μW. A 4-cell panel connecting a 1-cell panel produces 74.5 μW, and a 3-cell panel connecting a 2-cell panel produces 71.9 μW, while a 5-cell panel gives 77.1 μW. In contrast, the connection of two GBE panels will significantly decrease the overall output, from 20.0 μW for a 5-cell panel (Fig. 3b) to 7.4 μW for two connected 2-cell and 3-cell panels (Fig. S13f). The equivalence of electrical connection between multiple LBE panels should be ascribed to the minimized parasitic capacitance in each panel, whereas the

connection of two GBE panels significantly increases the overall parasitic capacitance. So, the LBE structure provides flexibility in designing and fabricating large-scale DEG panel arrays. In fact, the output power of our 6-, 7-, 8-, 9- and 10-cell panels in Fig. 3c is obtained from the electrical connection between 2- and 4-cell panels, 2- and 5-cell panels, 3- and 5-cell panels, 4- and 5-cell panels, and two 5-cell panels, respectively.

It is important to note the 5-cell LBE panel attains an overall average power of 77.1 μW at $f = 5$ Hz, which is 3.9 times higher than the 5-cell GBE panel at the same operation conditions (20.0 μW). It is also 2.5 times higher than the overall power (30.6 μW) of the best DEG panels reported in the literature[30]. Meanwhile, our LBE-DEG panels also have an energy harvesting efficiency $E_V$ (energy harvested per unit droplet volume) of 49.1 μJ mL$^{-1}$ (Fig. 3e), 4.2 times higher than the one in the literature (12.2 μJ mL$^{-1}$, Table S1)[30]. It is possible to further increase the overall output power through increasing the cell number in the panels (Fig. 3c) or increasing the working frequency for all the DEG cells (Fig. S17). However, due to the strong electrical interference in both cases, the energy harvesting efficiency $E_V$ from the droplets is significantly reduced, from 49.1 μJ mL$^{-1}$ for 5-cell panels at 5 Hz to 34.4 μJ mL$^{-1}$ for 10-cell panels at 5 Hz, 32.4 μJ mL$^{-1}$ for 5-cell panels at 8 Hz, and 23.4 μJ mL$^{-1}$ for 10-cell panels at 8 Hz (Fig. S17).

Moreover, we also compared the output between the 5-cell LBE panel operating at 5 Hz (Fig. 3e) and one single LBE-DEG cell operating at a high frequency of 25 Hz (Fig. 3f), as they consume comparable droplet volume. The former attains an average power of 77.1 μW, 1.8 times higher than the latter (41.8 μW). This superiority is attributed to the fact that, at high frequency, water droplets cannot maintain their structural integrity and tend to break into smaller droplets due to the high Weber number[38]. A smaller droplet volume results in a reduced spreading area, which in turn decreases the value of $C_B$ and leads to a lower peak output voltage (Eq. (1)). Although increasing the droplet frequency can increase the output power, from 32.7 μW at 5 Hz to 41.8 μW at 25 Hz, the energy harvesting efficiency $E_V$ decreases significantly, dropping from 93.4 μJ mL$^{-1}$ to 34.8 μJ mL$^{-1}$. Further increasing the frequency may cause more performance degradation due to the increased interference between the subsequent droplets. The strong interference makes it challenging for DEGs to harvest energy based on high-frequency droplets. This also explains the low energy harvesting efficiency $E_V$ of only 32.4 μJ mL$^{-1}$ in the ultrahigh-frequency (>165 Hz) DEGs[25] in spite of their high power density of 1257 mW m$^{-2}$ (Table S1). Taking into account the trade-off between overall output power, energy harvesting efficiency, and fabrication complexity, the 5-cell LBE-DEG panels at 5 Hz are selected for integration of large-scale DEG arrays in the next section.

**Integrating generator panels into generator arrays**
With the integration of full-wave rectifiers, our LBE panels can be integrated to form large-scale DEG arrays with minimized energy loss. As shown in Fig. 4a, 6 LBE-DEG panels were placed on the same board with tilted angle of 45°. All of them were rectified and connected in parallel. Driven by 30 droplet generators at the impinging frequency of 5 Hz, all panels generate similar output voltages (Fig. 4b), indicating the performance scalability of the LBE-DEG panels. The overall output of the 6 panels (or in total 30 cells) exhibits strongly irregular voltage peaks, most of which vary between 100 V and 400 V (Fig. 1b). We also evaluated the output performance of the DEG panels and the entire 30-cell DEG arrays under varying external load resistance. As shown in Fig. S9c, d, the maximum average power of one DEG panel (5 cells) and the DEG array (30-cell) reaches 85.9 μW and 371.8 μW, respectively. This implies a power degradation of only ~28% when the panel number increases from 1 to 6. Figure 4c indicates the dependence of $P_{RMS}$ on the panel number. When the panel number is no more than 3, the output power increases in proportion to the panel number, suggesting excellent performance scaling. The integration of more than 3 panels

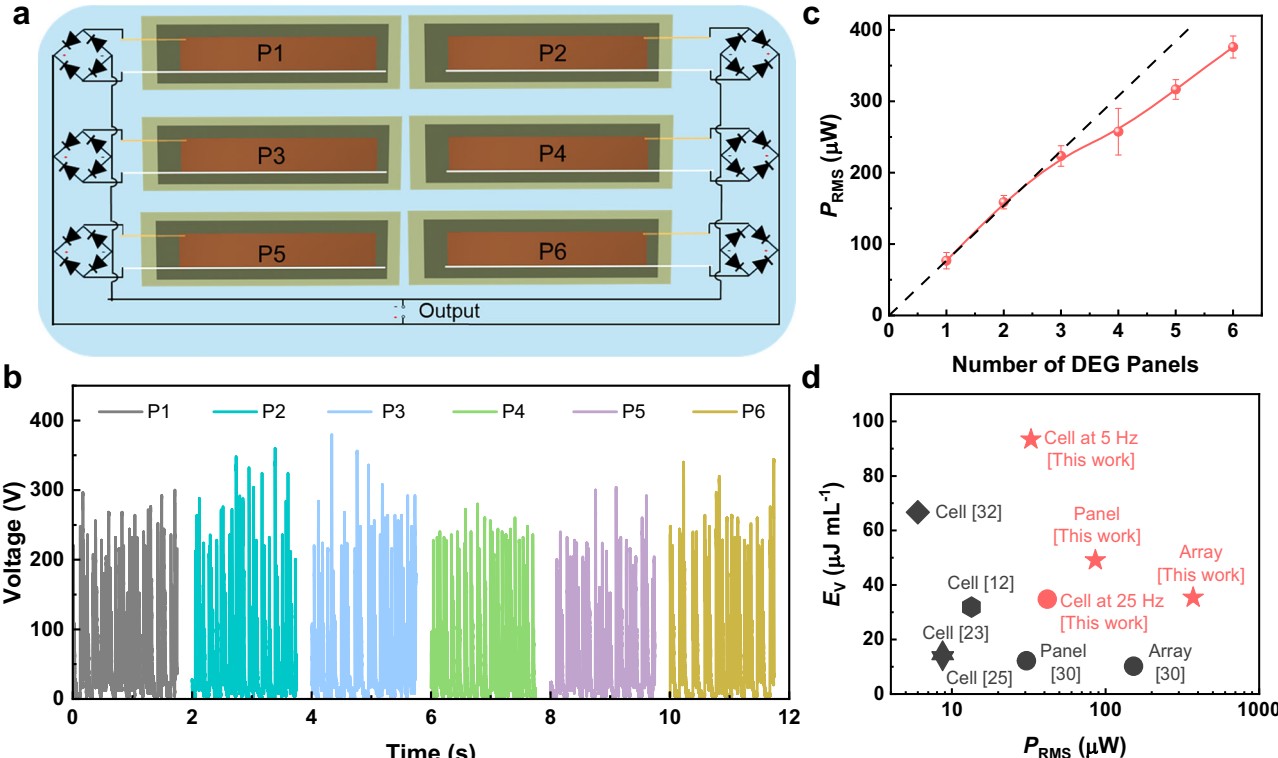

**Fig. 4 | Design and output performance of DEG panel arrays. a** Schematic illustration of a DEG panel array consisting of 6 LBE-DEG panels labeled as P$i$ ($i$ = 1, 2, …, 6). **b** Voltage-time curves of the 6 individual panels, each consisting of 5 DEG cells. **c** Dependence of $P_{RMS}$ of 30-cell (5 cell per panel × 6 panels) DEG panel array on the number of panels. Error bars represent the standard sample deviation. **d** Comparison of $E_V$ and $P_{RMS}$ among various DEG cells, panels and panel arrays[12,23,25,30,32]. Source data are provided as a Source Data file.

induces visible performance degradation. Since every panel has been rectified (Fig. 4a), the overall performance degradation of the DEG array should likely result from inter-panel parasitic capacitance, rather than the electrical interference. Nevertheless, as compared with other DEG cells, panels and arrays reported recently[12,23,25,30,32], our DEG arrays, exhibit significantly higher average output power and energy harvesting efficiency (Fig. 4d and Table S1). The overall average power of our 30-cell DEG array has exceeded 370 μW, more than twice higher than that (~150 μW) of the 30-cell DEG array in the literature[30]. The overall energy harvesting efficiency $E_V$ (35.4 μJ mL$^{-1}$) is even >3 times higher than the latter (10.2 μJ mL$^{-1}$), and comparable to that of most single DEG cells in the literature (Fig. 4d, Table S1).

**Performance of microsupercapacitor arrays and the integration**
The large-scale (30-cell) DEG arrays produce strongly irregular instantaneous high-voltage electricity (Fig. 1b). The strong irregularity makes it challenging for the present techniques to effectively store such electricity to form stable power to supply electronics[32,33]. So far, capacitors are often employed to store the output electricity of DEG arrays, but the energy storage efficiency is as low as <2%[29,30]. In our study, we also used a commercial capacitor with a capacitance of 470 μF to store the electricity generated by the 30-cell DEG array (Fig. S18). The capacitor was charged to 0.2 V within approximately 30 s and stored an energy of merely 9.9 μJ. The ESE, defined by the ratio of the energy stored in the energy storage component to the maximum energy produced in the energy harvesting component, is less than 1%. The low ESE should be attributed to the low voltage attained in the capacitors, typically <3 V. In theory, the stored energy is $E_C = 1/2$ $CV^2 = 1/2$ $QV$. Within a certain charging time $t$ that is much longer than

the period of DEG output pulses, the energy harvested by the DEGs is constant, and the charge $Q$ transferred from the DEGs to the capacitors could also be supposed to be constant. As $Q = CV$, small capacitance $C$ could lead to high voltage $V$, and hence large energy $E_C$ and high ESE. Accordingly, the working voltage window of the energy storage devices should be high enough to ensure being charged up to high $V$. However, the solution of high $V$-induced high ESE may not be favored by most electronics because they usually prefer low voltage and high current. To solve this problem, capacitor arrays are often used in TENG-based self-charging power systems with the strategy of "charging in series and discharging in parallel"[32]. During the charging process, the capacitors (each of capacitance $C$ and working voltage window of $\Delta V$) are connected in series so that the overall capacitance is reduced to $C/m$ with $m$ being the number of capacitors while the overall working voltage window increases to $m\Delta V$. The reduced overall capacitance and increased overall working voltage windows improve the ESE of storing energy from the TENGs. During the discharging process, the capacitors are connected in parallel to lower the output voltage and increase the current, so as to improve compatibility with general electronics.

However, capacitors are usually bulky. The integration of hundreds of capacitors in an array will lead to large form factors. In the literature[39–41], a variety of techniques and materials have been developed to fabricate large-scale MSC arrays with cell number up to 340 and working voltage windows up to 200 V (Fig. 5e). Such MSC arrays are insufficient to match our 30-cell DEG arrays due to the irregular high-voltage pulsed output ranging between 100 V and 400 V (Fig. 1b). In order to fabricate MSCs arrays with larger cell numbers and wider working voltage window, we extend our previous research[32] to

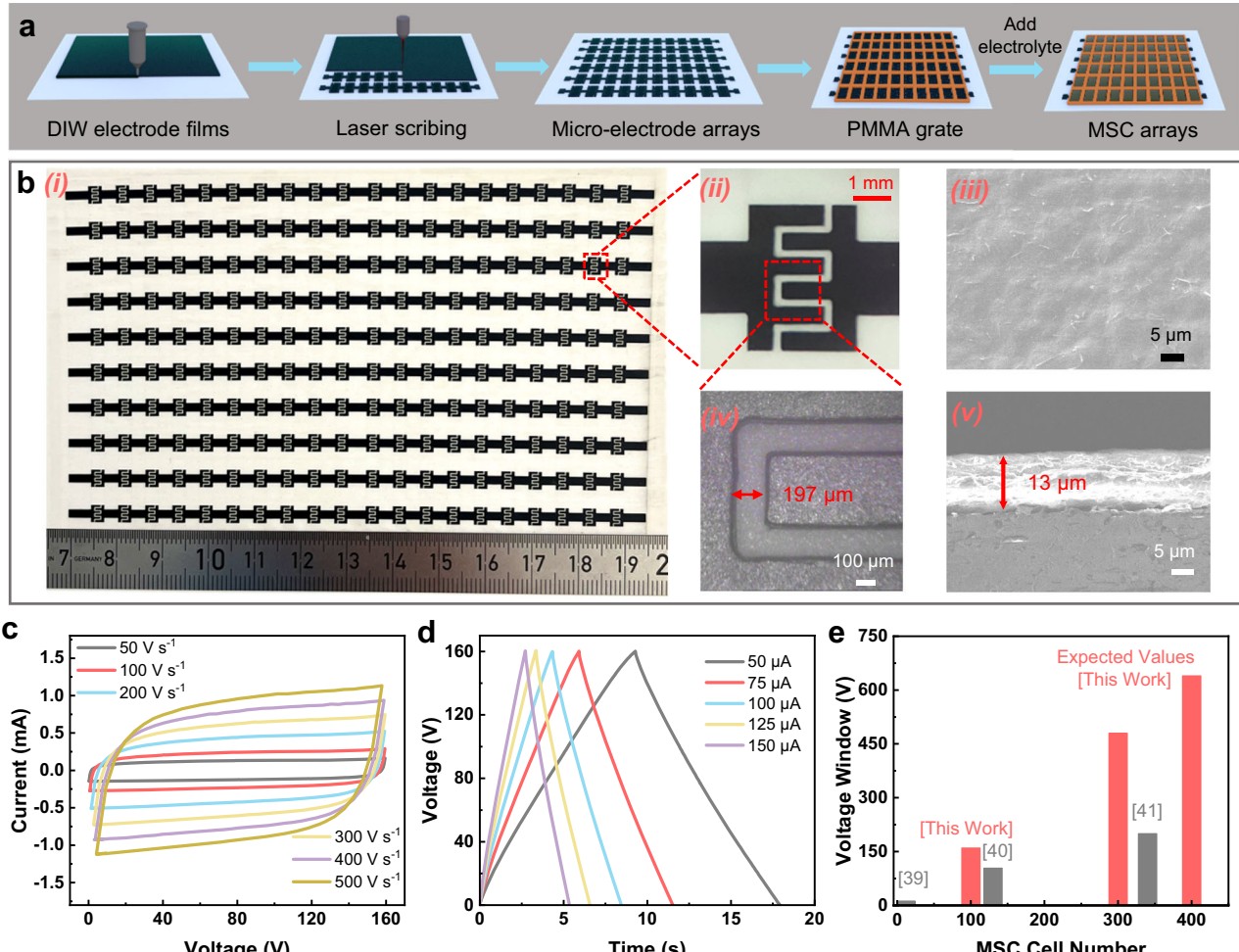

**Fig. 5 | Fabrication process and electrochemical measurement of MSC arrays.** **a** Schematic illustration of the fabrication process of MSC arrays (DIW: direct ink writing, PMMA: poly(methyl methacrylate)). **b** Photograph of electrodes of (i) MSC arrays and (ii, iv) single MSC cells, and scanning electron microscope (SEM) images of the poly(3,4-ethylenedioxythiophene):poly(styrene sulfonate) (PEDOT:PSS)-based electrodes in (iii) top view and (v) cross-sectional view. **c** CV curves of a 100-cell MSC sub-array at different scan rates. **d** Galvanostatic charge/discharge (GCD) curves of a 100-cell MSC sub-array at different currents. **e** Overall working voltage window versus cell number of various MSC arrays in this work and in recent literature[39–41]. Source data are provided as a Source Data file.

combine direct ink writing (DIW) of highly conductive metal-free organic inks with laser scribing (Fig. 5a) to reliably fabricate 400-cell MSC arrays on ceramic substrates. The conductive organic inks mainly comprise doped-conducting polymer poly(3,4-ethylenedioxythiophene): poly (styrene sulfonate) (PEDOT:PSS) and electrochemically exfoliated graphene (EEG). As validated in our previous research[32], the doped PEDOT:PSS is an excellent electrode material in which the co-existence of electronically conductive PEDOT networks and ionically conductive PSS networks allows individual MSCs to attain thickness-independent volumetric capacitance of ~5 F cm$^{-3}$ at high rate of 1 V s$^{-1}$ for electrode thickness up to 130 μm and working voltage window up to 1.6 V. The addition of graphene improves the ink stability without sacrificing the conductivity, as so to enable the formulation of high-concentration (up to 3 wt%) inks with viscosity >10$^3$ Pa s. In this work, the stable PEDOT:PSS/graphene inks with high viscosity and concentration were employed to efficiently and reliably print through DIW large-area (12.5 cm × 7.5 cm) uniform electrode films (Fig. 5b(i, iii)) with thickness >10 μm per printing pass (Fig. 5b(v)). The printed films were subsequently patterned into 20 rows of MSCs (Fig. 5b(i)) through laser scribing. Each row consists of 20 cells connected in series and each cell is of interdigitated structure with a footprint area of 3.0 mm × 4.0 mm (Fig. 5b(ii)). The two ends of each row are later connected with switches through copper tapes and silver pastes so that the 20 rows can be switched between series connection

and parallel connection to readily implement the principle of "charging in series and discharging in parallel". In conclusion, not only is the combination between direct ink writing and laser scribing a scalable method for efficient fabrication of the 400-cell MSC arrays of large electrode thickness >10 μm, but also is it a high-resolution method to enable reliable fabrication of small-size interdigitated MSCs with an inter-finger gap of merely ~200 μm (Fig. 5b(iv)). The narrow gaps enhance the transport of electrolyte ions, thereby contributing to the high-rate capability[42] of the MSCs.

In order to ensure the isolation of the electrolytes between the neighboring cells in the large-scale MSC array, a sheet of poly(methyl methacrylate) (PMMA) template (Fig. 5a) is attached with acid-resistant tape to cover the entire MSC array with an opening window in every cell position to fully expose the MSC electrodes. With the addition of gel electrolytes—comprising a mixture of poly (4-styrenesulfonic acid) (PSSH) and phosphoric acid (H$_3$PO$_4$)—into every opening window to bridge the electrodes, the MSC arrays are ready for electrochemical characterization (Fig. S19a, b). Since the working voltage window of the 400-cell arrays is far beyond the voltage limit of our electrochemical characterization equipment, cyclic voltammetry (CV), galvanostatic charge/discharge (GCD), and electrochemical impedance spectroscopy (EIS) were performed on four sub-arrays, each consisting of 100 cells (or 5 rows) connected in series. The key performance of the entire (400-cell) array is estimated as the equivalent performance of the

series connection of all the four sub-arrays. Specifically, the rate capability and working voltage window of the 400-cell array are the sums of that of all the 100-cell sub-arrays, while the overall capacitance of the array is the equivalent capacitance of series-connected sub-arrays. As shown in Fig. 5c, the CV curves of the 100-cell sub-arrays exhibit nearly rectangular shapes at various scan rates (including an ultrahigh scan rate of 500 V s$^{-1}$) within a voltage window of 160 V. The GCD curves display ideal symmetric triangular shapes with negligible dynamic voltage (IR) drop, even at currents up to 150 μA (Fig. 5d), indicating low equivalent series resistance (ESR) of the sub-arrays. This observation is further supported by EIS measurements (Fig. S19c, d), which reveal an ESR of ~5.6 kΩ. As shown in Fig. S20, after 2000 cycles of charge-discharge of a 100-cell MSC array at the full-scale working voltage window between 0 and 160 V (on average the voltage window per cell is 1.6 V), the capacitance remains around 78%. When the working voltage window per cell is narrowed to 1.0 V, a 30-cell MSC array can maintain 97% capacitance after 5000 cycles of charge/discharge between 0 and 30 V. Notably, when the MSC arrays are charged by the DEGs, the maximum working voltage is far below the full scale, and one may expect much smaller capacitance degradation after a large number of cycle times. Moreover, the 100-cell sub-arrays demonstrate high overall capacitance, with a total capacitance of 2.4 μF at a scan rate of 50 V s$^{-1}$ and 1.6 μF at 500 V s$^{-1}$ as calculated from the CV curves (Fig. S21a). The GCD measurements indicate a capacitance of 2.7 μF at a charging/discharging current of 50 μA, retaining 95% of this value when the current is increased to 150 μA (Fig. S21b). Consistent results were obtained from three other 100-cell sub-arrays (Fig. S22), underscoring the reliability and scalability of our fabrication technique for large-scale MSC arrays. According to the scaling laws of series-connected capacitors, our entire 400-cell MSC arrays can attain an overall capacitance of ~0.4 μF at the scan rate of 2000 V s$^{-1}$ and working voltage window of 640 V. It has significantly outperformed the reported MSC arrays in the literature (Fig. 5e). More importantly, the ultrawide working voltage window of 640 V and ultrahigh rate capability of 2000 V s$^{-1}$ make the 400-cell MSC arrays competent to store the strongly irregular pulsed electricity generated by the 30-cell DEG arrays with peak voltage > 400 V (Fig. 1b), as discussed in the next section.

**Charging large-scale microsupercapacitor arrays with large-scale generator arrays**
We have constructed an SCPS where the 30-cell DEG array is used to charge the 400-cell MSC array (Figs. 6a, S23a). Following the principle of "charging in series and discharging in parallel"[43], our MSC arrays were charged using different numbers of cells in series and then discharged in a combined parallel-series configuration, denoted as "$m_p P \times m_s S$" (where $m_p$ represents the number of parallel-connected groups, and $m_s$ represents the number of series-connected cells within each group). For instance, the configuration "5P × 20S" indicates charging across 100 series-connected MSC cells, followed by discharging in parallel connection of 5 groups, each containing 20 series-connected MSC cells. A series of switches were used to transfer between series connection and combination connection of the MSC array (Fig. 6a, series connection is realized through simply switching on all $S_{s,i}$ switches and switching off all $S_{pa,i}$ and $S_{pb,i}$ switches ($i = 1, 2, ..., 9$); conversely, parallel connection is through switching on all $S_{pa,i}$ and $S_{pb,i}$ switches and switching off all $S_{s,i}$ switches). As shown in Fig. S23b, c, the MSC arrays, with varying cell numbers, were charged to 12 V by the 30-cell DEG array. The charging time decreases drastically with increasing the MSC cell number (Fig. S24a), suggesting higher charging rates and hence higher power for the larger-scale MSC arrays. To explore the efficiency of our SCPS, MSC arrays with a variety of "$m_p P \times m_s S$" configurations were charged by the 30-cell DEG array for a fixed duration of 30 s, followed by discharging at a constant current of 5 μA (Fig. 6b

and Fig. S23d, e) for the calculation of stored energy in the MSCs. In spite of the same charging time, the MSC arrays with larger cell number exhibit longer discharging time and higher discharging voltage (Fig. 6b). As shown in Fig. 6c, the stored energy increased significantly with the cell number, reaching the maximum of 2.44 mJ for the 400-cell array (discharged at 10P × 40S configuration). The ESE[44] of our SCPS, i.e., discharge energy (2.44 mJ) of the MSCs divided by the maximum output energy of the DEGs (371.8 μW × 30 s), reaches 21.8%. In view of the strongly irregular high-voltage (maximum peak >400 V) energy supply by the 30-cell DEG array (Fig. 1b) and our chipless SCPS system, such an ESE has already gone far above the best performance of similar SCPSs with only single or a few capacitors in the literature (Fig. 1d). As shown in Fig. 1d and Table S1, the similar SCPS[30] comprising 30-cell DEGs and one single normal capacitor can only reach an ESE of 2.0%. The SCPS[25] comprising one single DEG of high frequency (165 Hz) DEG and hence irregular low-voltage (peak voltage <200 V) energy supply can only get an ESE of 7.5%. Without the assistance of commercial chips, the SCPS[33] comprising a TENG with irregular high-voltage output can only reach an ESE of 0.3%. Besides, we conducted similar tests to charge a 200-cell MSC array, respectively, with one single DEG cell operating at 5 Hz, 1 cell at 25 Hz, and one DEG panel consisting of 5 cells at 5 Hz. As shown in Fig. S25, ESE remains ~11% for the 1 cell at 25 Hz and 5 cells at 5 Hz but decreases to ~6% for 1 cell at 5 Hz (Table S2). The low ESE for the 1 cell at 5 Hz may be attributed to the mismatch between the DEG output and the MSC arrays. At 5 Hz, the DEG produces a peak voltage exceeding 400 V (Fig. S25a), which is significantly higher than the maximum working voltage window (320 V) of the 200-cell MSC arrays. In contrast, the peak voltage generated by 1 cell at 25 Hz (Fig. S25b) and 5 cells at 5 Hz (Fig. S25c) remains within this window, resulting in similar ESE in both cases. This highlights the critical role of the extended voltage window in enhancing the ESE of our large-scale MSC arrays. Furthermore, when the charging time varies within the range between 20 s and 40 s (Fig. S26), the stored energy increases substantially with the cell number as well as the charging time, while the output power is primarily determined by the cell number rather than the charging time (Fig. S26f). This confirms the advantage of the large-scale MSC arrays in increasing the ESE.

To gain more insight into the advantage of large-scale MSCs in the SCPS, we further investigated the dependence of the average stored energy per individual MSC cell, $E_{cell}$, on the cell number (Fig. S24b). With increasing the cell number up to 400, $E_{cell}$ first increases linearly and then becomes to saturate, with the transition at the cell number of ~150. At the linear region, $E_{cell}$ increases with the total cell number, implying that the same MSC cell in a large-scale array can store more energy than in a small-scale array. This advantage should be mainly ascribed to the significantly increased voltage window and charging rate in the large-scale MSC arrays, both of which are beneficial for improving the efficiency in storing pulsed high-voltage electricity[32]. Once the cell number exceeds 150, the working voltage window of the MSC array reaches >240 V, close to the peak voltage of the DEG panels (Fig. 3e) and arrays (Fig. 1b). This could be the reason for the saturation of $E_{cell}$. Additional research is necessary to unveil the accurate mechanism. Nevertheless, the total stored energy for the same charging time still increases continuously and nearly linearly with the cell number throughout the range studied in this work (up to 400 cells, Fig. 6c). Benefitting from both large-scale (30-cell) DEG arrays and large-scale (400-cell) MSC arrays, the effective energy harvesting power of our SCPS (discharge energy of MSC arrays divided by charging time of the DEG arrays) has reached 81.2 μW, 27 times higher than that (3.0 μW, Fig. 6d) of the best SCPS comprising a 30-cell DEG array in the literature[30]. It is also 15 times higher than that (5.4 μW) of our previous SCPS[32] comprising one single DEG and 90-cell MSC arrays. These suggest the effectiveness and importance of scaling both DEGs and MSCs for practical water energy harvesting.

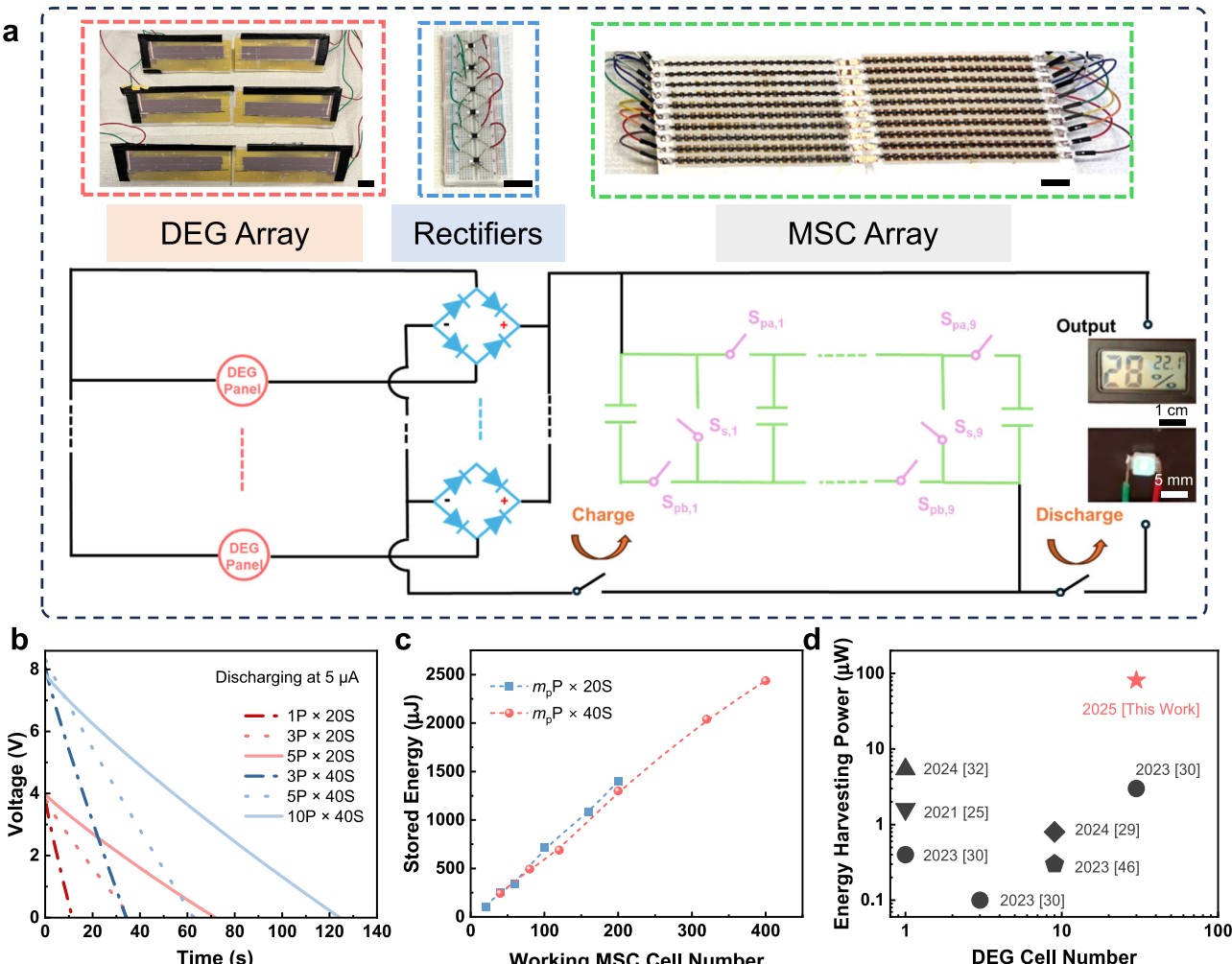

**Fig. 6 | SCPS based on integration of large-scale DEG arrays and large-scale MSC arrays. a** Photographs (upper) of the key components (left: a 30-cell DEG array, middle: 6 rectifiers, right: a 400-cell MSC array) in our SCPS and the circuit design (lower). $S_{s,i}$, $S_{pa,i}$, and $S_{pb,i}$ ($i = 1, 2, ..., 9$) are the switches to control the connection configuration of the MSC arrays. Unmarked scale bars are 2 cm. **b** Discharging curves at a current of 5 μA of MSC arrays with different configurations of "$m_p$P × $m_s$S" after being charged by the 30-cell DEG array for 30 s, where $m_p$ represents the number of parallel-connected groups, and $m_s$ represents the number of series-connected cells within each group. **c** Energy stored in the MSC arrays with different numbers of working cells after being charged by the 30-cell DEG array for 30 s calculated from the discharging curves, some of which are shown in (**b**). **d** Comparison of energy harvesting power (stored energy in the energy storage components divided by the DEG charging time) of various DEG-based SCPSs between this work and the literature[25,29,30,32,46]. Source data are provided as a Source Data file.

Finally, we demonstrated the practical applications of our SCPS in powering common electronic devices. With a charging time of 30 s, the 200-cell MSC arrays were able to continuously light an LED (Cree, XLamp, MX-6 LEDs) for 35 s (in the discharging configuration of 10P × 20S, Video S1), while the 400-cell MSC arrays (in the discharging configuration of 10P × 40S) extended this duration to 60 s (Video S2). After a charging period of 30 s, the 200-cell MSC arrays could power a hygrometer (INF Company AB, Mini Liquid Crystal Display Hygrometer) continuously for approximately 40 s (Video S3). Moreover, natural water sources, such as rainwater, can also be used to drive our DEG panel arrays to enhance their practical applicability. We have measured the output performance of a 30-cell DEG panel array driven by simulated rainwater (the lab-prepared raindrop-mimicking liquid with composition listed in Table S3). As shown in Fig. S27, despite the somewhat decreased performance as compared with DI water, the DEG panel arrays with rainwater are still able to charge MSC arrays and power electronic devices. Specifically, after being charged by the 30-cell DEG panel array with rainwater for 30 s, the 200-cell MSC array stored over 950 μJ of energy (as compared with 1250 μJ for DI water) and were able to light an LED for around 25 s (Video S4) and power a calculator for around 70 s (Video S5).

## Discussion

In conclusion, we have developed a chipless self-charging power system comprising large-scale DEG arrays integrated with large-scale MSC arrays to effectively harvest water energy. It has been found that the global bottom electrodes of the present DEGs are an important yet ignored factor that prevents the performance improvement of individual DEG cells and accounts for the significant performance degradation in the upscaled DEG arrays. Simply through localizing the bottom electrode to an area comparable to the droplet spread area, the adverse parasitic capacitance related to the DEG cells or panels is nearly minimized while the beneficial bulk capacitance is still close to the maximum, resulting in a record average output power of 32.7 μW for individual DEG cells, nearly 50% higher than the best value in literature. The localized bottom electrodes also allow to build single-rectifier DEG panels with up to 5 DEG cells sharing one single rectifier to obtain scalable average output power of 85.9 μW. Then 6 DEG panels are integrated into large-scale (30-cell) DEG arrays to obtain an average output power of 371.8 μW, more than twice as high as the 30-cell DEG array in the literature. In order to effectively store the strongly irregular high-voltage (> 400 V) output electricity of the large-scale

DEG arrays, compatible large-scale MSC arrays have been fabricated through the combination between DIW and laser scribing to integrate up to 400 cells and obtain expected working voltage window of 640 V and charging rate of 2000 V s$^{-1}$. It has been validated that the 400-cell MSC array can store the output electricity of 30-cell DEG arrays at the efficiency of 21.8%. The high energy storage efficiency increases the power of our 30-cell DEG array-based water energy harvesting system to a record value of 81.2 μW, 27 times higher than that of the 30-cell DEG system in the literature. All these prove the promise to integrate large-scale DEG arrays with large-scale MSC arrays for practical clean energy harvesting from natural water.

## Methods

### Fabrication of droplet-based electricity generator cells, panels and arrays

For each DEG cell, a PTFE film (Walfront, 150 μm thick) was cut into 5 cm × 7 cm and cleaned with acetone, isopropanol, and deionized water. Then a conductive copper tape with different sizes (according to the LBE or GBE design) was pasted on one side of the PTFE film as the bottom electrode. An aluminum tape was pasted on the other side as the top electrode. Each electrode was connected to a copper wire for electrical tests. The whole device was fixed on a 5 cm × 7 cm glass slide with a double-side Kapton tape. The DEG panels were fabricated on PMMA substrates in a similar way to the cells, just with increased electrode size. The PTFE film and PMMA substrate had the same size of 15 cm × 6 cm. Multiple panels were connected to form an array, where each panel was equipped with a full-wave rectifier (MDB10S, Fairchild Semiconductor International, Inc.).

### Fabrication of the microsupercapacitor arrays

The functional PEDOT:PSS ink was formulated in a similar way to our previous reports[32,45]. In brief, around 40 mg EEG was mixed with 8 mL PEDOT:PSS (1.1 wt% water dispersion, Product No. 739332, Sigma-Aldrich) and 2 mL ethylene glycol (99.5%, Product No. 1.09621, Sigma-Aldrich). The dispersion was sonicated for 30 min and stirred for 1 h. Then, the ink was dried under vacuum at room temperature overnight to evaporate a part of the ink solvents and increase the viscosity to a suitable value for DIW. With this ink, a rectangular electrode film was printed in one pass on an alumina plate (CERcuits, 190 × 140 × 1 mm$^3$) through the FELIX BIO printer (FELIXprinters) equipped with a disposable 10 mL syringe (Fisherbrand, nozzle diameter 400 μm) at the substrate temperature of 80 °C. Then the electrode film was scribed into interdigitated electrode arrays with a pulsed fiber laser (GoldMark, 30 W power, λ = 1064 nm). To avoid short circuit between the electrolytes of neighboring cells, one sheet of grate was prepared. The double-sided tape (3 M GPT-020F) was first attached to a cleaned PMMA plate (Darenyi, 250 × 200 × 1 mm$^3$), and then scribed by CO$_2$ laser (Universal Laser Inc., 30 W power, λ = 10.6 μm) into the desired grid pattern (a sheet with open windows for separating electrolytes between neighboring MSC cells, as shown in Fig. 5a). After the grate was attached to the MSC electrode arrays, a gel electrolyte was cast into every grid to bridge the interdigitated electrodes of the MSC cell. The electrolyte was prepared by mixing 0.5 mL PSSH solution (Mw ≈ 75 000, 18 wt.% in H$_2$O, Sigma-Aldrich) with 0.14 mL H$_3$PO$_4$ (≥85%, Sigma-Aldrich).

### Characterization and measurement of droplet-based electricity generators

A disposable infusion set (Evercare Medical) comprising a flow regulator, a plastic nozzle, and a plastic tube was used as a droplet generator. The flow rate could be controlled through the flow regulator. The spacing between droplets could be controlled by adjusting the spatial layout of infusion sets. The droplet volume could be adjusted by tailoring the inner diameter of the plastic nozzle connecting to the plastic tube. If not specified, we measured all the DEGs under the same conditions: deionized water droplets of 70 μL impinge onto the 45°-

tilted DEG plane from a height of 30 cm and at a frequency of 5 Hz under indoor conditions, with the temperature and relative humidity maintained at 22 °C and 28%, respectively.

The roughness of the PTFE film and glass slide was measured by Atomic force microscopy (AFM) (NanoScope Dimension 3100, Veeco/Digital Instruments). The device parasitic capacitance was measured between the top and bottom electrodes via an RCL meter (PM 6303, Philips). The output voltage of the DEG cells, panels and arrays was measured directly by using oscilloscope (RSDS 1152CML + , RSPRO, Sweden) with a high-impedance ($R_{tip}$ = 100 MΩ) probe in the absence of external load resistors. The following equations were used to calculate the average voltage $U_{RMS}$, and the average power $P_{RMS}$ of the DEG cells/panels/arrays,

$$U_{RMS} = \sqrt{\frac{\int_0^T U(t)^2 dt}{T}} \qquad (2)$$

$$P_{RMS} = \frac{U_{RMS}^2}{R_{tip}} \qquad (3)$$

where $U(t)$, $T$, and $R_{tip}$ = 100 MΩ refer to the measured voltage over time, the total time of the measurement, and the probe resistance of the oscilloscope, respectively. In the presence of external load resistors, the voltage divider method was used to measure $U(t)$, as shown in Fig. S9a where the calculation of the average current and power is specified.

All the DEG measurements have been repeated across multiple devices (at least 2 devices for each measurement). Each of the reported datapoints, such as peak voltage and average power/voltage, is the average over ten statistical samples. Each sample for calculating the average power/voltage is a section extracted from a tested voltage-time curve with a duration of around 0.5–3.5 s. All the error bars of the reported data are the standard sample deviation.

### Characterization of microsupercapacitor arrays

The surface morphology of the printed PEDOT:PSS electrodes was characterized by SEM (Gemini Ultra 55, Zeiss, Germany). For the electrochemical tests of the MSC arrays, CV and GCD were conducted via a Keithley 4200A-SCS parameter analyzer (Tektronix, Inc.), and EIS was carried out in a two-electrode system using an electrochemical working station Gamry Interface 1010E (Gamry Instruments Inc., Warminster PA, USA). Before the test, silver paste and copper tape were applied to the two leads of the MSC arrays for external connection.

The capacitance was calculated based on the CV or GCD curves. The following equation is used for the CV curves,

$$C_{CV} = \frac{\int_0^{\Delta V} (I_C - I_D) dV}{2 \nu \Delta V} \qquad (4)$$

where $I_C$, $I_D$, $\Delta V$, and ν refer to the charging current, discharging current, voltage window, and scanning rate, respectively.

The following equation is used for the GCD curves,

$$C_{GCD} = \frac{I_D \Delta t}{\Delta V} \qquad (5)$$

where $\Delta t$ is the discharging time.

When the MSC arrays were charged by the DEG panel arrays, the overall stored energy was calculated from the following equation $E_{stored} = \frac{1}{2} I_D \Delta V \Delta t$.

### Finite element method simulation

The FEM simulation was performed using the ANSYS Electronics software, with a 3D DEG model (Fig. S28) consisting of a sheet of

dielectric material (PTFE or glass, of lateral dimension 7 cm × 5 cm for all LBE-DEGs and increased to 15 cm × 15 cm for the GBE-DEG whose bottom electrode area is set to 15 cm × 12 cm) sandwiched in between the top and bottom electrodes (Copper, both of thickness 20 μm). The thickness of PTFE and glass was set to 150 μm and 1.0 mm, respectively. For the calculation of $C_B$, the top electrode was a circular copper electrode with a diameter of 2 cm to mimic the spread droplets, locating with the center aligned to the bottom electrode (Fig. S28c). For the calculation of $C_{P,D}$, the top electrode was a rectangular copper electrode with dimension of 5 cm × 0.2 cm, locating over the bottom electrode with no overlapping (Fig. S28b). In both cases, the bottom electrode was a rectangle with varying sizes (from 0.5 cm × 0.5 cm to 15 cm × 12 cm). To simplify the boundary condition setting, the DEG was placed at the center of a large vacuum box of dimensions 30 cm × 30 cm × 20 cm (Fig. S28a).

The potential distribution in the model is described by the Poisson equation:

$$\nabla^2 \Phi = -\frac{\rho}{\varepsilon_r \varepsilon_0}$$

where $\Phi$ is the electric potential, $\varepsilon_0$ is the permittivity of vacuum, $\varepsilon_r$ is the relative permittivity of the medium ($\varepsilon_r = 2.1$ for PTFE, 5.5 for glass and 1.0 for vacuum) and $\rho$ is the volume charge density ($\rho = 0$ everywhere in our simulations). The surfaces of the top electrode and bottom electrode were assigned with a fixed surface charge of 1 pC and -1 pC, respectively. The natural Neumann boundary condition $\partial \Phi / \partial n = 0$ is applied to the outer surfaces of the vacuum box where $n$ denotes the normal to the boundary. After the FEM meshing and simulation, the capacitance was calculated through dividing the assigned charge on the electrode surfaces by the potential difference between the top and bottom electrodes. Besides, for simplicity, some capacitance was calculated directly through the Q3D Extractor, a specialized solver within Ansys Electronics Desktop for extracting parasitic parameters, including capacitance. Both methods gave almost the same results.

## Data availability
All data supporting the findings of this study are provided within the article and the Supplementary Information file. Source data are provided with this paper.

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

## Acknowledgements

We thank Prof. Carl-Mikael Zetterling at KTH for many constructive discussions during the period of manuscript revision. We acknowledge the financial support from the Swedish Research Council (Grant No. 2019-04731, J.L.), the Swedish Energy Agency (Grant No. P2023-00800, J.L.), and the Swedish Foundation for International Cooperation in Research and Higher Education (STINT, CH2017-7284, J.L.).

## Author contributions

Z.L. and J.L. conceived the experiment, data analysis, and interpretation. Z.L. performed the fabrication of devices. Z.L. and S.C. carried out the synthesis of materials. Z.L., S.C., and Y.F. performed the measurement and characterization of devices. Z.L. and J.L. conducted the simulations. The manuscript was written by Z.L. and J.L. The project was planned, directed, and supervised by J.L. All authors discussed the results and commented on the manuscript.

## Funding

## Competing interests

The authors declare no competing interests.
