## [Transparent Peer Review file · Nature Communications]

Efficiency Optimization for Large-Scale Droplet-Based Electricity Generator Arrays with Integrated Microsupercapacitor Arrays

Corresponding Author: Professor Jiantong Li

Version 0:

Reviewer comments:

Reviewer #1

(Remarks to the Author)

The authors present a study on the development of large-scale droplet-based electricity generator (DEG) arrays integrated with microsupercapacitor (MSC) arrays for efficient water energy harvesting. The manuscript explores voltage and power signals as well as energy storage efficiency in DEG systems, aiming to optimize device performance through localized bottom electrodes (LBE) and improve energy storage through high-voltage MSC arrays.

However, the novelty of this study appears to be insufficient for publication in Nature Communications. A major concern lies in the reliability of the performance evaluation, which is a key strength of this work. Specifically, the quantification of device dimensions and droplet dynamics is lacking, making it difficult to rigorously compare the unit cell performance and array-scale integration with previous studies. Without a thorough parameterization of droplet impact conditions (e.g., velocity, size distribution, impact angle) and device architecture (e.g., active area per cell, scaling effects), the reported improvements in output power and energy storage efficiency remain unconvincing.

For a future submission, the authors should improve the overall clarity of the manuscript by providing more quantitative explanations, particularly in the methodology and performance comparison sections. Addressing the above concerns with more rigorous experimental validation and comparative analysis would significantly strengthen the study.

Given these limitations, I do not recommend publication of this manuscript in Nature Communications in its current form.

Reviewer #2

(Remarks to the Author)

This manuscript presents a chipless self-charging power system that combines large-scale droplet-based electricity generator arrays with large-scale microsupercapacitor arrays to efficiently harvest water energy. The performance of these devices is impressive, and I recommend this work for publication. However, I have a few comments:

1. Why did the authors use PTFE? Does it have any advantages over other polymer materials?
2. Please clearly explain the charge transfer mechanism when the droplets come into contact with the top electrode.

Reviewer #3

(Remarks to the Author)

The manuscript "Large Scale Droplet based Electricity Generator Arrays Integrated with Large Scale Microsupercapacitor Arrays for High Efficiency Water Energy Harvesting" reports an experimentally detailed study on the development of a large-scale droplet-based electricity generator array, optimized by localizing bottom electrodes and its integration with a 400-cell microsupercapacitor array for efficient water energy harvesting. The manuscript addresses critical challenges in DEG scalability, panel-level parasitic capacitance, and energy storage inefficiency when dealing with high-voltage pulsed outputs. It is therefore my view that this manuscript has the interest to be published in Nature Communications, although major revisions should be performed.

-The submitted manuscript mentions an improved stability of the LBE-DEGs, but the authors only performed 3-hour tests. Have longer tests (days or weeks) been conducted under ambient or humid conditions to validate stability claims? Is the array sensitive to temperature or environmental variations?

-The behavior of the droplet impact is the key for the DEG operation. However, the effects of droplet breakup or instability at high frequencies (e.g. 25 Hz) are not detailed. Could the authors further elaborate on the limitations imposed by droplet behavior and how performance varies with water composition (e.g., ion concentration)?

-Why does circuit capacitance depends on SBE? Why does it first decreases and then increases with SBE (Figure 2e)?

-Given the large number of devices and their integration aspects, the manuscript should enhance its statistical analysis of the data. Have measurements been repeated across multiple devices or test cycles? Please provide standard deviations to support the reproducibility of the results.

-How where the numerical simulations performed? Are these 2D or 3D, what are the boundary conditions?

-How sensitive is the energy storage efficiency to variations in droplet frequency and size? Is the system robust to naturally fluctuating water sources (e.g., rainfall, wave splashes)?

-Can the MSC arrays be recharged repeatedly without degradation? If so, how many cycles were tested, and is there any evidence of degradation?

-Please specify how droplet volume, spacing and flow rate are controlled and whether these parameters impact inter-cell interference or power variations.

Reviewer #4

(Remarks to the Author)

This paper presents the optimization of the DEG system composition for the aspect of the size of the bottom electrode to enhance the energy harvesting, and its utilization as power source for charging micro-supercapacitor array. With this novel approach, energy generated from the DEG can be successfully increased, and power storage in the micro-supercapacitor can be effective for the DEG application for power source of various electronic devices, thus the hurdle which the DEG faces on its practical utilization can be successfully solved. Especially, approaches for the DEG optimization are appropriate for showing its necessity. In this context, I think this research can be published after significant improvement by answering major and minor comments written below.

- 1) First of all, I think the connection between the DEG optimization for large scale energy harvesting and its integration to the large-scale micro-supercapacitor array is weak right now. Why the DEG is suitable for power storage in the micro-supercapacitor? Does the bottom electrode tailoring allows micro-supercapacitor charging with DEG? At this moment, it is difficult to find the imperativeness of this integration, and I think this weak connection seems to obscure the novelty of this study. I recommend to author to add it in the manuscript, for this manuscript to be suitable for the publication.
- 2) In Figure 2e, it is shown that the CP,C suddenly decreases 10 times, and then rapidly increases when the SBE and SD,max are similar, which is distinct from the CB and CP,D behavior. What will be the reason of this sudden change? The author should state its reason for the clarity of the manuscript, because it is the important experimental data for the strategy. Also, I think the author needs to state the detailed methodology for the measuring the parasitic capacitance for the readers.
- 3) Although the electrical circuit model of the DEG is dominant for the DEG optimization in this manuscript, type of the water utilized for the model analysis and output generation is not specified. Considering that the electrical conductivity of the water highly affects the equivalent circuit modeling and electrical output generation behavior, I think the author should state about the specification of the water droplet for the completeness of the manuscript.
- 4) In Figure 3e and 3f, the author shows the distinct output behavior from the 5-cell at 5 Hz case and 1-cell at 25 Hz case. For the 5-cell at 5 Hz case, it shows the uneven output generation from 100 V to 400 V which is resulted from the constructive and destructive interference. However, for the 1-cell at 25 Hz case, it shows relatively even output generation. As the author calculated the average power of them, 5-cell at 5 Hz case generates around 1.4 times higher electrical output than 1-cell at 25 Hz case. To supplement this difference, I think author should add the capacitor charging behavior under these two cases for emphasizing this deviation, because it is one of the important experiments for the novelty.
- 5) In Figure S4, the maximum average power under the DEG cell, DEG panel, and DEG array is shown. First of all, I think the author needs to make legend in the graph. Right now, it is hard to clarify the experimental data due to absence of the legend. Furthermore, it seems that the optimal resistance, where the load resistance under highest average power, is higher than the previous researches, which shows the hundreds of k Ω to 30 M Ω . What will be the reason of this distinct behavior on it? Finally, it seems that the optimal resistance is decreasing, as the platform is changed from DEG cell to DEG array. What should be the reason of this behavior? The reference which the author can check is written below:
 - [1] Jang, Sunmin, et al. "Beyond metallic electrode: spontaneous formation of fluidic electrodes from operational liquid in highly functional droplet-based electricity generator." *Advanced Materials* 36.35 (2024): 2403090.
 - [2] Zhang, Nan, et al. "A droplet-based electricity generator with ultrahigh instantaneous output and short charging time." *Droplet* 1.1 (2022): 56-64.
 - [3] Zhang, Nan, et al. "Performance transition in droplet-based electricity generator with optimized top electrode arrangements." *Nano Energy* 106 (2023): 108111.
 - [4] Wang, Lili, et al. "Monolithic integrated flexible yet robust droplet-based electricity generator." *Advanced Functional Materials* 32.49 (2022): 2206705.
- 6) In Figure S5, it seems that the output is decreased as the droplet continuously impinges for 3 hours. However, in the general knowledge, the electrical output requires to be increased with the surface charging resulted from the continuous liquid-solid contact electrification along the DEG operation, as shown in previous study (Xu, Wanghuai, et al. "A droplet-based electricity generator with high instantaneous power density." *Nature* 578.7795 (2020): 392-396.). What would be the reason of this output degradation behavior?
- 7) In Figure 4, the author shows the DEG array composed of 6 DEG panel, and it seems that the RMS power is increasing as the number of DEG panels increases. What will happen on CP,C and CP,D, as the size of the DEG panel increases and number of DEG panels increases?

8) For the micro-supercapacitor (MSC) array charging demonstrated in Figure 6, how long will it take the MSC to be charged to the level for practical utilization with single DEG array?

9) Furthermore, it seems that the MSC array with 400 cells possess lower capacitance than the commercial capacitor, which are widely utilized for the energy harvesting researches. What will be the advantageous aspect of using MSC array rather than the commercial capacitor for the aspect of the practical utilization? Also, why the MSC array in this manuscript possesses slow discharging behavior even under the low capacitance? I think the readers who are not familiar with the supercapacitor will have difficulties to understand this manuscript and experimental data right now. The author must provide kind and detailed explanations on each data for the readers' understanding.

10) Generally, for the spontaneous operation of the DEG, the target condition should involve rainfall. Considering that raindrops contain various ions and chemical components, they will inevitably affect the conductivity of the droplets, leading to changes in the output generation behavior. In this context, I believe the author should demonstrate the MSC charging behavior using rainwater (or a lab-prepared raindrop-mimicking liquid) and evaluate the electronic device operation under these conditions, as this paper focuses on the practical utilization of the DEG.

11) It shows typos in the Figure of supplementary information. The author needs to check overall manuscript to avoid typos.

Version 1:

Reviewer comments:

Reviewer #1

(Remarks to the Author)

Reviewer #2

(Remarks to the Author)

The authors have adequately addressed all comments, and I recommend the manuscript for acceptance in its current form.

Reviewer #3

(Remarks to the Author)

The revised version of the manuscript answered all the raised questions satisfactorily and is therefore ready to be published.

Reviewer #4

(Remarks to the Author)

The revised manuscript is ready for its acceptance.

Response to Reviewers' Reports

Dear Reviewers:

Thank you very much for carefully reviewing our manuscript. We are truly appreciative of the valuable comments from the Reviewers, which have significantly contributed to improving the quality of our work. In response to these suggestions, we have conducted additional experiments to investigate the influence of liquid properties and droplet dynamics on the performance of LBE-optimized DEGs. Moreover, we have conducted several charging tests of the MSC arrays to further demonstrate the practical applicability of our self-charging power system.

We have thoroughly addressed all the comments and revised the manuscript accordingly. Below is our point-by-point response to every comment from the reviewers. All the revision is tracked in both the manuscript and the supplementary information. **All the page and line numbers mentioned in this Response refer to the revised manuscript with markup.** We sincerely look forward to further suggestions from the reviewers and hope that the revised manuscript meets the high standards of *Nature Communications*.

Point-by-Point Response:

Reviewer #1 (Remarks to the Author):

The authors present a study on the development of large-scale droplet-based electricity generator (DEG) arrays integrated with microsupercapacitor (MSC) arrays for efficient water energy harvesting. The manuscript explores voltage and power signals as well as energy storage efficiency in DEG systems, aiming to optimize device performance through localized bottom electrodes (LBE) and improve energy storage through high-voltage MSC arrays.

However, the novelty of this study appears to be insufficient for publication in *Nature Communications*. A major concern lies in the reliability of the performance evaluation, which is a key strength of this work. Specifically, the quantification of device dimensions and droplet dynamics is lacking, making it difficult to rigorously compare the unit cell performance and array-scale integration with previous studies. Without a thorough parameterization of droplet impact conditions (e.g., velocity, size distribution, impact angle)

and device architecture (e.g., active area per cell, scaling effects), the reported improvements in output power and energy storage efficiency remain unconvincing.

For a future submission, the authors should improve the overall clarity of the manuscript by providing more quantitative explanations, particularly in the methodology and performance comparison sections. Addressing the above concerns with more rigorous experimental validation and comparative analysis would significantly strengthen the study.

Given these limitations, I do not recommend publication of this manuscript in Nature Communications in its current form.

Response: Thank you very much for pointing out the flaw in validating the advantages of our LBE-DEGs.

We fully agree with the reviewer that the DEG performance depends on a number of parameters concerning droplet dynamics and device architecture/dimension. In addition, different parameters may play different roles in the final DEG performance. It should be challenging to straightforwardly compare in a qualitative manner our LBE-optimized DEG performance with those GBE-DEGs reported in the literature that in general have different droplet dynamics and device architecture. To make our comparison as fair as possible, we make efforts from the following five aspects.

- (1) We have included three key performance indicators (KPIs): E_V (the harvested energy per unit volume of water), \hat{P}_{RMS} (average output power density), and **ESE** (energy storage efficiency, the ratio of the energy stored in the energy storage component to the *maximum* energy produced in the energy harvesting component). On the one hand, the KPIs are able to characterize the *absolute performance* of the DEGs for various applications. On the other hand, the impact of droplet dynamics and device architecture has been taken into account in the KPIs. For example, E_V measures the harvested energy per unit volume of water, so that the effects of different droplet size, frequency, velocity, and impact angle have all been normalized, at least to some extent. The average output power \hat{P}_{RMS} has also normalized the effects of droplet dynamics and device dimension.
- (2) For all the literature data used for comparison, we *only cite the highest/optimized values* reported in the publication *under similar test conditions*, as shown in **Table S1** where almost all the droplet size is within a range of 50-100 μL , impinging height (equivalent to

velocity) within 15-30 cm, impinging frequency around 5-10 Hz, impact angle between 15° and 45°, and maximum droplet spreading area around 3 cm².

- (3) Without LBE optimization, our GBE-DEGs do not evidently outperform those reported in the literature. It can be inferred that the high performance of our LBE-DEGs really results from the LBE optimization, not other factors.
- (4) In the revision, we have validated that the LBE optimization of the DEG performance applies to *all test conditions* (e.g., different droplet size, water type, impinging height, angle, and environmental temperature and humidity, etc.). For details, please refer to **Figures S2-S6** and our response to Comments 1 and 2 of Reviewer #3. This suggests that the LBE effects are an intrinsic factor to influence/determine DEG performance.
- (5) A detailed analysis of scaling effects is provided in the original manuscript, as now shown on **Page 12 Line 24 to Page 14 Line 15** in the revision, to validate that upscaling from individual cells to panels induces significantly less output performance degradation in our LBE-DEGs than the general GBE-DEGs.

Based on the discussion above, we believe it is credible to conclude that our LBE-DEGs have improved the output performance of the general GBE-DEGs, in terms of E_V , \hat{P}_{RMS} , and ESE, as shown in Figures 1d,4d, and Table S1. To clarify our points, in the revision we have included a comprehensive set of test conditions in **Table S1** and stressed on **Page 11 Lines 28-31** that our LBE-DEGs are tested with similar droplet dynamics and device architecture to the general GBE-DEGs in the literature.

Devices	Water Type	Droplet Volume (μL)	Droplet Frequency	Impact Angle ($^\circ$)	Droplet Height (cm)	Droplet Spreading Area (cm^2)	U_{RMS} (V)	P_{RMS} (μW)	E_V ($\mu\text{J mL}^{-1}$)	\hat{P}_{RMS} (mW m^{-2}) ^a	ESE	Energy Harvesting Power (μW)	Ref
Cell	Tap Water	100	4.2 Hz	15	15	2.7	11.6	13.4 ^b	31.9	49.4	-	-	12
Cell	Tap Water	60	1.5 Hz	45	25	2.6	13.5	6.0 ^b	66.7	27.5	-	-	23
Cell	Tap Water	4	165.0 Hz	45	-	0.17	14.6	21.4 ^b	32.4	1256.6	7.5%	1.6	25
Cell	DI Water	50	9.0 Hz	35	10	-	25.8	8.7	14.9	8.7	62%	5.4	32
Cell	Tap Water	53	-	45	20	2.5	-	-	-	-	-	-	29
Cell	Tap Water	72	30 mL min ⁻¹ (6.9 Hz)	30	16	2.2	-	-	-	-	-	0.4	30
Cell	DI Water	70	5 Hz	45	30	3.0	54.8	32.7	93.4	109.0	-	-	This Work
Panel	Tap Water	53	-	45	20	-	-	-	-	-	-	0.8	29
Panel	Tap Water	99.5	5.1 Hz	-	15	-	-	-	-	-	-	0.3	45
Panel	Tap Water	72	3 Droplet at 50 mL min ⁻¹ (11.6 Hz)	30	30	2.2	6.3	30.6	12.2	46.4	0.6%	0.2	30
Panel	DI Water	70	5 Droplet at 5 Hz	45	30	3.0	86.0	85.9	49.1	57.3	-	-	This Work
Panel Arrays	Tap Water	72	30 Droplets at 20-30 mL min ⁻¹ (6.9 Hz)	30	30	2.2	50.5	152.2	10.2	23.1	2.0%	3.0	30
Panel Arrays	DI Water	70	30 Droplet at 5 Hz	45	30	3.0	177.3	371.8	35.4	41.3	21.8%	81.2	This Work

^a Average power density. ^b Results are calculated from the product of the generated electrical energy per droplet with the impinging frequency, reported in papers.

Supplementary Table 1. Comparison of performance among various DEG devices.

Supplementary Figure 3| Output performance of DEG cells of various bottom electrode area S_{BE} driven by droplets from various falling heights. a-c Output voltage of the DEG cells with droplet falling heights of (a) 10 cm, (b) 30 cm, and (c) 40 cm. **d** Photographs of the droplets with different falling heights just spreading to the maximum area on the PTFE film.

Supplementary Figure 4| Output performance of DEG cells of various bottom electrode area S_{BE} driven by droplets with various impact angles. a-c Output voltage of the DEG cells with droplet impact angles of (a) 15° , (b) 45° , and (c) 75° . **d** Photographs of the droplets with different impact angles just spreading to the maximum area on the PTFE film.

Supplementary Figure 5| Output performance of DEG cells of various bottom electrode area S_{BE} driven by droplets of various volumes. a-d Output voltage of the DEG cells with t droplet volumes of (a) 26 μL , (b) 48 μL , (c) 70 μL , and (d) 98 μL . The droplet volume is controlled by the inner diameter of the nozzles in the droplet generators. In (a-d), the inner nozzle diameters are 1.8 mm, 2.2 mm, 3.4 mm, and 4.5 mm, respectively. **e** Photographs of the droplets with different volumes just spreading to the maximum area on the PTFE film.

Reviewer #2 (Remarks to the Author):

This manuscript presents a chipless self-charging power system that combines large-scale droplet-based electricity generator arrays with large-scale microsupercapacitor arrays to efficiently harvest water energy. The performance of these devices is impressive, and I recommend this work for publication. However, I have a few comments:

1. Why did the authors use PTFE? Does it have any advantages over other polymer materials?

Response: Thank you very much for your relevant comments.

PTFE is a very common and popular triboelectric material for the fabrication of high-performance TENGs/DEGs, including the first DEGs [Xu, W. et al. A droplet-based electricity generator with high instantaneous power density. *Nature* **578**, 392-396 (2020)], thanks to its many unique advantages, such as very high surface charge density, strong hydrophobicity, good stability and low cost.

2. Please clearly explain the charge transfer mechanism when the droplets come into contact with the top electrode.

Response:

The DEGs have been invented since 2020. As a matter of fact, their working mechanism was already well explained in the first paper on DEGs [Xu, W. et al. A droplet-based electricity generator with high instantaneous power density. *Nature* **578**, 392-396 (2020)]. In addition, a follow-up [Li, Luxian, et al. Sparking potential over 1200 V by a falling water droplet. *Science Advances* **9**, eadi2993 (2023)] has complemented the theory into a comprehensive circuit model to describe the charge transfer mechanism. In the manuscript, we only employ these well-accepted frameworks to briefly explain the charge transfer mechanism as the basic principles for optimizing our LBE-enhanced DEGs, as explained in original manuscript, now shown on **Page 6 Lines 14-21** in revision: “Prior to droplet impact, the negatively charged PTFE induces a positive charge on the bottom electrode. At a certain time (t_1 in Figure S1), the impinging droplet spreads to its maximum area and contacts the top electrode to quickly transfer the positive charge from the bottom electrode to the top electrode and generate a high instantaneous voltage. During the retraction of the droplet, the positive charge gradually returns to the bottom electrode, resulting in a transition of the DEG output from positive to negative voltage. Finally, the droplet detaches from the top electrodes (at the time of t_3 in Figure S1), and both the charge and output voltage drop to zero¹². ”

If the readers/reviewers are interested in the comprehensive charge transfer mechanism of DEGs, it is important to refer to the relevant publications, such as *Nature* 2020 (Ref. 12) and *Science Advances* 2023 (Ref. 23).

Reviewer #3 (Remarks to the Author):

The manuscript “Large Scale Droplet based Electricity Generator Arrays Integrated with Large Scale Microsupercapacitor Arrays for High Efficiency Water Energy Harvesting” reports an experimentally detailed study on the development of a large-scale droplet-based electricity generator array, optimized by localizing bottom electrodes and its integration with a 400-cell microsupercapacitor array for efficient water energy harvesting. The manuscript addresses critical challenges in DEG scalability, panel-level parasitic capacitance, and energy storage inefficiency when dealing with high-voltage pulsed outputs. It is therefore my view

that this manuscript has the interest to be published in Nature Communications, although major revisions should be performed.

(1) the submitted manuscript mentions an improved stability of the LBE-DEGs, but the authors only performed 3-hour tests. Have longer tests (days or weeks) been conducted under ambient or humid conditions to validate stability claims? Is the array sensitive to temperature or environmental variations?

Response: Thank you very much for your very valuable comments.

We have now extended the test time to 8 hours using two different water sources (deionized water and tap water), as shown in **Figure S10**. These continuous 8-hour test time has already been far longer than all the tests in the existing DEG reports so far. Our tests suggest that DEG exhibits good stability when using deionized water, with only ~30% performance degradation after 8 hours. However, the degradation should be mainly attributed to the accumulation of moisture and droplet residues on the PTFE surface, which reduces the interaction area between the sequent droplets and the PTFE surface. The residue issue is more severe in the case of tap water. Nevertheless, **the DEG performance can be almost fully recovered** simply by using paper tissue to clean the PTFE surface. This implies that the performance degradation is not any intrinsic failure of the DEG devices, but just the alteration of the operating conditions (moisture and droplet residues accumulated on PTFE surface) due to long-time interaction between PTFE surfaces and impinging droplets. So, we believe *with proper maintenance* the DEGs have great potential to work for weeks or even longer. We have now updated the long-term tests and relevant discussion on **Page 10 Line 31 to Page 11 Line 7**.

Supplementary Figure 10 | Long-term stability test of one LBE-DEG cell. **a, c** Output voltage-time curves of the DEG cell driven by (a) DI water and (c) tap water. In each case, after a test time of 8 hours, the DEG cell was cleaned though wiping with paper tissue. **b, d** Time-resolved average power P_{RMS} and power retention of the DEG cell driven by (b) DI water and (d) tap water.

The influence of test environment on DEG performance has already been studied in previous research. In general, high temperature and high humidity reduce the DEG performance due to the accumulation of more moisture (similar to the conditions after long-time operation). For comprehensive knowledge of the influence, please refer to [Figure 4a in *Nature* **578**, 392–396 (2020); Figure 2h in *Adv. Mater.* **35**, 2209713 (2023)]. Nevertheless, in this work, following the reviewer’s suggestion, we have validated our claimed LBE-area effects on DEG performance under different environmental conditions. Due to the lack of equipment to well control humidity and temperature in our lab, we tested our LBE-DEGs with deionized water under both indoor (Temperature: 22 °C, Relative Humidity: 23%) and outdoor (Temperature: 10 °C, Relative Humidity: 92%, on a rainy day in May 2025 in Stockholm) environments. As shown in **Figure S2**, with increasing the relative humidity from 23% to 92%, the DEG cells only retained ~60% output power, in agreement with other studies. Similar environment-induced performance degradation should also apply in DEG panels and panel arrays. However, under all the test environments (as well as with different droplet dynamics as in **Figures S3-S5**), the DEG performance is always optimal when the LBE area is close to the maximum droplet spreading area. This implies that the DEG

performance depends intrinsically on the LBE area. In other words, our strategy of LBE optimization is adaptable for all DEG designs. We have added the relevant discussion on Page 7, Lines 8-15.

Supplementary Figure 2| Outdoor test of DEG cells with different bottom electrode area S_{BE} driven by DI water. **a** Photograph of one DEG cell operated outdoors (Temperature: 10 °C, Relative Humidity: 92%). **b,c** Output voltage of the DEG cells tested under (b) outdoor and (c) indoor environments.

(2) The behavior of the droplet impact is the key for the DEG operation. However, the effects of droplet breakup or instability at high frequencies (e.g. 25 Hz) are not detailed. Could the authors further elaborate on the limitations imposed by droplet behavior and how performance varies with water composition (e.g., ion concentration)?

Response: Thank you very much for your very insightful comments.

In general, too high a frequency will decrease the DEG efficiency due to the reduced droplet volume and increased inter-droplet interference. At high frequencies, the droplet cannot maintain a large volume and tends to split into smaller droplets. A smaller droplet volume results in a reduced spreading area, which in turn decreases the value of C_B and leads to a lower peak output voltage (Equation 1). Although increasing the droplet frequency can increase the output power, from 32.7 μW at 5 Hz to 41.8 μW at 25 Hz, the energy harvesting efficiency E_V (harvested energy per unit water volume) decreases significantly, dropping from 93.4 $\mu\text{J mL}^{-1}$ to 34.8 $\mu\text{J mL}^{-1}$. Further increasing the frequency may cause more performance degradation due to the increased interference between the subsequent droplets. The strong interference, as also discussed in [Wang, L. et al. Harvesting energy from high-frequency impinging water droplets by a droplet-based electricity generator. *EcoMat* **3**, e12116 (2021)], makes it challenging for DEGs to harvest energy based on high-frequency droplets. We have added the relevant discussion on Page15, Lines 20-27.

We also investigated the impact of different liquids and ion concentrations on DEG performance, with results presented in **Figure S6**. The increase in concentration of ions and chemicals in water tends to degrade the output performance of the DEGs, which is generally consistent with previous reports^{23,29}. However, it is important to note that regardless of the type of water used, the LBE effects remain in all the cases (**Figure S2-S5**), i.e., the DEG performance is always optimal at $S_{BE} = S_{D,max}$. We have added the relevant discussion in the manuscript (**Page 7, Lines 8-15**).

Supplementary Figure 6| Output performance of DEG cells of various bottom electrode area S_{BE} driven by various types of water. a-c Output voltage of the DEG cells driven by (a) tap water, (b) rainwater, and (c) seawater. **d-f** Output voltage of the DEG cells driven by aqueous NaCl solutions of different concentrations: (d) 0.1 mol L⁻¹, (e) 0.3 mol L⁻¹, and (f) 0.5 mol L⁻¹.

(3) Why does circuit capacitance depends on SBE? Why does it first decreases and then increases with SBE (Figure 2e)?

Response: Thank you very much for pointing out this critical issue.

After careful consideration, we found the first two fitting values of C_{PC} in Figure 2e were not accurate. We extracted C_{PC} from the fitting of Eq. (S3) to the experimentally measured V_{peak} with a series of external parallel-connected capacitors C_E . In our original manuscript, during the fitting we put more weight on the first datapoint, i.e., $V_{peak}(C_E = 0)$. It gives good overall fitting for all the cases when $S > S_{BE}$. But for $S < S_{BE}$, the overall fitting is much poorer (as shown in our **original Figure S3b-c** as copied below). In the revision, we

put equal weight on all datapoints of $V_{\text{peak}}(C_E)$ to improve the overall goodness of fit, as shown in the **new Figure S8b-c** (as also copied below). In the new fitting, C_{PC} monotonically increases with S_{BE} , as shown in the **new Figure 2e**. The new data of C_{PC} should be physically more reasonable. We updated the discussion on **Page 10, Lines 16-18**. We apologize for the suboptimal data fitting in our original manuscript.

Original Supplementary Figure 3| Measurement of the circuit-induced parasitic capacitance. b-c Peak voltage dependence on C_E for the DEG cell when $S < S_{\text{BE}}$.

Revised Supplementary Figure 8| Measurement of the circuit-induced parasitic capacitance. b-c Peak voltage dependence on C_E for the DEG cell when $S < S_{\text{BE}}$.

Figure 2 | e Dependence of the bulk capacitance and parasitic capacitance on S_{BE} .

We have added the description to detail the fitting model in **Supplementary Note 1**: “The parasitic circuit capacitance $C_{P,C}$ is measured according to the protocol in the previous research²³. As illustrated in Figure S8a, an external variable capacitor C_E is connected between the top and bottom electrodes of the DEGs so that C_E is parallel with C_P . One may then replace C_P in Equation (S2) with $C'_P = C_P + C_E$ and obtain the C_E -dependent peak voltage V_{peak} as

$$V_{\text{peak}} = \frac{C_B U_0}{C_B + C'_P} = \frac{Q}{C_B + C_{P,D} + C_{P,C} + C_E} = \frac{\sigma_S S_{\text{eff}}}{C_B + C_{P,D} + C_{P,C} + C_E}. \quad (\text{S3})$$

For a DEG with specified structure, C_B and $C_{P,D}$ are fixed while $Q = C_B U_0 = \sigma_S S_{\text{eff}}$ is also fixed with σ_S being the permanent surface charge density on the PTFE film and $S_{\text{eff}} = \min(S_{BE}, S_{D,\text{max}})$ being the minimal value between the BE area and the maximum droplet spread area. Then one can measure a series of V_{peak} with varying C_E and extract the value of $C_{P,C}$ from Equation (S3). In this work, for each DEG with S_{BE} ranging from 1 cm² to 180 cm², we vary C_E from 0 to 1000 pF to measure V_{peak} . Equation (S3) is used to fit the experimental data with $\sigma_S = 48 \mu\text{C}/\text{m}^2$ and extract the values of $C_{P,C}$ (Figure 2e). The surface charge of density of PTFE film $\sigma_S = 48 \mu\text{C}/\text{m}^2$ used in our fitting is comparable to the value of $\sim 35 \mu\text{C}/\text{m}^2$ in the previous research²³.”

However, the mechanism for the variation of $C_{P,C}$ with S_{BE} has not been clear to us. It should result from the interaction between the test circuit and the top/bottom electrodes. Similar dependence of $C_{P,C}$ on the **top electrode** area in GBE-DEGs was also found in the paper [Li, Luxian, et al. Sparking potential over 1200 V by a falling water droplet. *Science Advances* **9**, eadi2993 (2023)]. But an in-depth mechanism deserves systematic exploration in the future.

(4) Given the large number of devices and their integration aspects, the manuscript should enhance its statistical analysis of the data. Have measurements been repeated across multiple devices or test cycles? Please provide standard deviations to support the reproducibility of the results.

Response: Thank you very much for your very careful comments.

We have added the explanation for the statistics of our reported data on **Page 28 Line 3-8**. “All the DEG measurements have been repeated across multiple devices (at least 2 devices for each measurement). Each of the reported datapoints, such as peak voltage and average power/voltage, is average over ten statistical samples. Each sample for calculating the average power/voltage is a section extracted from a tested voltage-time curve with a duration of around 0.5-3.5 seconds. All the error bars of the reported data are the standard sample deviation.”

In addition to these short test cycles, we also conducted long test cycles (8-hour tests) to further support the stability and reproducibility of our results.

(5) How where the numerical simulations performed? Are these 2D or 3D, what are the boundary conditions?

Response: Thank you very much for your very careful comments.

We have updated the relevant discussion on **Page 29, Lines 4-29**: “The finite element method (FEM) simulation was performed using the ANSYS Electronics software, with a 3D DEG model (Figure S28) consisting of a sheet of dielectric materials (PTFE or glass, of lateral dimension 7 cm × 5 cm for all LBE-DEGs and increased to 15 cm × 15 cm for the GBE-DEG whose bottom electrode area is set to 15 cm × 12 cm) sandwiched in between the top and bottom electrodes (Copper, both of thickness 20 μm). The thickness of PTFE and glass were set to 150 μm and 1.0 mm, respectively. For the calculation of C_B , the top electrode was a circular copper electrode with a diameter of 2 cm to mimic the spread droplets, locating with the center aligned to the bottom electrode (Figure S28c). For the calculation of $C_{P,D}$, the top electrode was a rectangular copper electrode with dimension of 5 cm × 0.2 cm, locating over the bottom electrode with no overlapping (Figure S28b). In both cases, the bottom electrode was a rectangle with varying sizes (from 0.5 cm × 0.5 cm to 15 cm × 12 cm). To simplify the boundary condition setting, the DEG was placed at the center of a large vacuum box of dimensions 30 cm × 30 cm × 20 cm (Figure S28a).

The potential distribution in the model is described by the Poisson equation:

$$\nabla^2 \Phi = -\frac{\rho}{\epsilon_r \epsilon_0}$$

where Φ is the electric potential, ϵ_0 is the permittivity of vacuum, ϵ_r is the relative permittivity of the medium ($\epsilon_r = 2.1$ for PTFE, 5.5 for glass and 1.0 for vacuum) and ρ is the volume charge density ($\rho = 0$ everywhere in our simulations). The surfaces of the top electrode and bottom electrode were assigned with a fixed surface charge of 1 pC and -1 pC, respectively. The natural Neumann boundary condition $\frac{\partial \Phi}{\partial n} = 0$ is applied to the outer surfaces of the vacuum box where n denotes the normal to the boundary. After the FEM meshing and simulation, the capacitance was calculated through dividing the assigned charge on the electrode surfaces by the potential difference between the top and bottom electrodes. Besides, for simplicity, some capacitance was calculated directly through the Q3D Extractor, a specialized solver within Ansys Electronics Desktop for extracting parasitic parameters, including capacitance. Both methods give almost the same results.”

Supplementary Figure 28| Simulation model for capacitance calculation. a 3D view of the overall simulation model. The DEG cell is placed at the center of a large vacuum box. b,c Different (isometric, top and side) views for the simulation models of the DEG cells for calculating (b) $C_{P,D}$ and (c) C_B .

(6) How sensitive is the energy storage efficiency to variations in droplet frequency and size? Is the system robust to naturally fluctuating water sources (e.g., rainfall, wave splashes)?

Response: Thank you very much for your very valuable comments.

As noted in your Comment 2, the droplet frequency is strongly correlated to droplet size. Here, to investigate whether the energy storage efficiency (ESE) varies with droplet frequency and size, we conducted similar charging experiments using 200-cell MSC arrays

with the single DEG cell operating at 5 Hz (droplet size 70 μL) and 25 Hz (droplet size 48 μL), and a DEG panel consisting of 5 cells (droplet size 70 μL) at 5 Hz. As shown in **Figure S25**, ESE remains $\sim 11\%$ for the 1 cell at 25 Hz and 5 cells at 5 Hz but decreases to $\sim 6\%$ for 1 cell at 5 Hz (**Table S2**). This implies that during our tests, the ESE does not intrinsically vary with frequency and droplet size. The low ESE for the 1 cell at 5 Hz may be attributed to the mismatch between the DEG output and the MSC arrays. At 5 Hz, the DEG produces a peak voltage exceeding 400 V (**Figure S25a**), which is significantly higher than the maximum working voltage window (320 V) of the 200-cell MSC arrays. In contrast, the peak voltage generated by 1 cell at 25 Hz (**Figure S25b**) and 5 cells at 5 Hz (**Figure S25c**) remains within this window, resulting in similar ESE in both cases. We have added the relevant discussion on **Page 22, Lines 10-20**.

Supplementary Figure 25| Charge-discharge of 200-cell MSC arrays with different DEGs under different operating conditions. a-c Output voltage of the different DEGs (a) one DEG cell at 5 Hz, (b) one DEG cell at 25 Hz, and (c) one DEG panel consisting of 5 DEG cells at 5 Hz. **d** Discharge curves of the 200-cell MSC arrays at a configuration of $10\text{P} \times 20\text{S}$ under a current of 5 μA , after being charged for 30 s by the three different DEGs.

Regarding fluctuating water sources, usually one should not use them to directly drive DEGs. Instead, we recommend collecting such water in a tank as a buffer and using it to supply the DEG panel arrays via droplet generators. This approach ensures stable supply of droplets and constant output power. It is difficult to claim that our system is robust enough to withstand direct impacts from fluctuating water sources such as wave splashes. Certainly, it seems not necessary to do that either.

(7) Can the MSC arrays be recharged repeatedly without degradation? If so, how many cycles were tested, and is there any evidence of degradation?

Response: Thank you very much for your very valuable comments.

The MSC arrays can be recharged repeatedly, but not without any degradation. As shown in **Figure S20**, after 2000 cycles of charge-discharge of a 100-cell MSC array at the full-scale working voltage window between 0 and 160 V (on average the voltage window per cell is 1.6 V), the capacitance remains around 78%. When the working voltage window per cell is narrowed to 1.0 V, a 30-cell MSC array can maintain 97% capacitance after 5000 cycles of charge/discharge between 0 and 30 V. Notably, when the MSC arrays are charged by the DEGs, the maximum working voltage is far below the full scale, and one may expect much smaller capacitance degradation after a large number of cycle times. We have added the relevant discussion on **Page 20, Lines 21-28**.

Supplementary Figure 20| Long-cycle charge/discharge tests of MSC arrays. a Cycling test of a 100-cell MSC array for 2000 GCD cycles at a current of 150 μA within a voltage window of 160 V. **b** Cycling test of a 30-cell array for 5000 CV cycles at a scan rate of 30 V s⁻¹ within a voltage window of 30 V.

(8) Please specify how droplet volume, spacing and flow rate are controlled and whether these parameters impact inter-cell interference or power variations.

Response: Thank you very much for your very valuable comments.

Droplet volume, spacing, and flow rate are controlled using droplet generators. “A disposable infusion set (Evercare Medical) including a flow regulator, a plastic nozzle, and a plastic tube was used as the droplet generator. The flow rate could be controlled by using the flow regulator. The spacing between droplets could be controlled by adjusting the layout of infusion sets. The droplet volume could be adjusted by tailoring the inner diameter of the plastic nozzle connecting to the plastic tube.” (added on **Page 27, Lines 4-13**)

During the scale-up process from cells to panels, in order to optimize device performance, we employed our LBE design. Under this design, the droplet spreading area matches the bottom electrode area, while the spacing between droplets is minimized but also avoids the interference between the adjacent droplets. In other words, for an LBE-DEG panel, the optimal droplet volume and spacing should already be pre-designed. Any change to these two parameters will decrease the output power (due to suboptimal operation conditions for the cells), no matter whether there is inter-cell interference or not.

When the droplet volume and spacing are fixed, the flow rate can be considered as equivalent to the droplet frequency. Frequency variation can significantly affect the inter-cell interference and output power. With increasing frequency, the output power increases as expected in a proportional manner, but inter-cell interference also increases. The latter will hinder the power increase, which causes non-proportional increase of power with frequency. A more comprehensive discussion is provided in the original manuscript, now shown on **Page 12 Line 24 to Page 13 Line 16, and Page 15 Lines 3-32** in revision.

Thank you again for your very careful comments to help us improve the quality of this manuscript.

Reviewer #4 (Remarks to the Author):

This paper presents the optimization of the DEG system composition for the aspect of the size of the bottom electrode to enhance the energy harvesting, and its utilization as power source for charging micro-supercapacitor array. With this novel approach, energy generated from the DEG can be successfully increased, and power storage in the micro-supercapacitor

can be effective for the DEG application for power source of various electronic devices, thus the hurdle which the DEG faces on its practical utilization can be successfully solved. Especially, approaches for the DEG optimization are appropriate for showing its necessity. In this context, I think this research can be published after significant improvement by answering major and minor comments written below.

1) First of all, I think the connection between the DEG optimization for large scale energy harvesting and its integration to the large-scale micro-supercapacitor array is weak right now. Why the DEG is suitable for power storage in the micro-supercapacitor? Does the bottom electrode tailoring allows micro-supercapacitor charging with DEG? At this moment, it is difficult to find the imperativeness of this integration, and I think this weak connection seems to obscure the novelty of this study. I recommend to author to add it in the manuscript, for this manuscript to be suitable for the publication.

Response: Thank you very much for your very valuable comments.

In this work, the DEGs, particularly the tailored bottom electrode, are designed to improve output performance, but not intentionally to match the MSCs. Instead, our MSC arrays are developed to match the characteristics of the DEG panel arrays. The output signal from the DEG panel array (and most mechanical harvesters) is irregular high-voltage (up to 400 V) pulsed electricity which cannot be effectively (efficiency < 2%) stored by traditional energy storage devices (batteries, supercapacitors or capacitors) due to their low working voltage window (typically < 10 V) and/or slow charge rate (typically < 1 V/s for batteries and supercapacitors). In contrast, our large-scale MSC array has working voltage window up to 640 V and charging rate up to 2000 V/s. This merit makes our MSC arrays uniquely advantageous to store the irregular high voltage pulsed output electricity of DEG arrays with significantly improved efficiency (> 20%). We have a more detailed discussion in the original manuscript, now shown on **Page 17 Line 16 to Page 18 Line 10** in revision: “The large-scale (30-cell) DEG arrays produce strongly irregular instantaneous high-voltage electricity (Figure 1b). The strong irregularity makes it challenging for the present techniques to effectively store such electricity to form stable power to supply electronics. So far, capacitors are often employed to store the output electricity of DEG arrays, but the energy storage efficiency is as low as < 2%.

The low ESE should be attributed to the low voltage attained in the capacitors, typically < 3 V. In theory, the stored energy is $E_C = 1/2 CV^2 = 1/2 QV$. Within a certain charging time t much longer than the period T of DEG output pulses, the energy harvested by the DEGs is constant, and the charge Q transferred from the DEGs to the capacitors could

also be supposed to be constant. As $Q = CV$, small capacitance C could lead to high voltage V , and hence large energy Ec and high ESE. However, the solution of high V -induced high ESE may not be favored by most electronics because they usually prefer low voltage and high current. To solve this problem, capacitor arrays are often used in TENG-based self-charging power systems with the strategy of “charging in series and discharging in parallel”. During the charging process, the capacitors (each of capacitance C) are connected in series so that the overall capacitance is reduced to C/m with m being the number of capacitors. The reduced overall capacitance improves the ESE to store energy from the TENGs. During the discharging process, the capacitors are connected in parallel to lower the output voltage and increase the current, so as to improve compatibility with general electronics.”

2) In Figure 2e, it is shown that the $C_{P,C}$ suddenly decreases 10 times, and then rapidly increases when the SBE and SD_{max} are similar, which is distinct from the CB and $C_{P,D}$ behavior. What will be the reason of this sudden change? The author should state its reason for the clarity of the manuscript, because it is the important experimental data for the strategy. Also, I think the author needs to state the detailed methodology for the measuring the parasitic capacitance for the readers.

Response: Thank you very much for your very insightful comments.

Reviewer #3 shares the same concerns. To avoid repetitive responses, please refer to our response to Comment 3 of Reviewer #3.

The parasitic capacitance in our paper can be divided into device parasitic capacitance $C_{P,D}$, and circuit parasitic capacitance $C_{P,C}$. $C_{P,D}$ is measured directly with an RCL meter (PM 6303, Philips) by connecting both electrodes (**Page 27, Lines 18-19**). However, it's very hard to measure $C_{P,C}$ directly, so we use the fitting model (Equation S3) to extract the value of $C_{P,C}$. We have updated the discussion on **Page 10, Lines 16-18**, and **Supplementary Note 1**.

3) Although the electrical circuit model of the DEG is dominant for the DEG optimization in this manuscript, type of the water utilized for the model analysis and output generation is not specified. Considering that the electrical conductivity of the water highly affects the equivalent circuit modeling and electrical output generation behavior, I think the author should state about the specification of the water droplet for the completeness of the manuscript.

Response: Thank you very much for your very careful comments.

We apologize for missing the very important information in the original manuscript. Unless specified, all the droplets we used for experiments are deionized water. We have added relevant description on **Page 27, Lines 9-13**. “If not specified, we measured the DEG under the same conditions: deionized water droplets of 70 μL impinge onto the 45°-tilted DEG plane from a height of 30 cm and at a frequency of 5 Hz under indoor conditions, with the temperature and relative humidity maintained at 22 °C and 28%, respectively.” We also conducted additional experiments to investigate the dependence of DEG performance on different water types. For details, please refer to our response to Comment 2 of Reviewer #3 .

4) In Figure 3e and 3f, the author shows the distinct output behavior from the 5-cell at 5 Hz case and 1-cell at 25 Hz case. For the 5-cell at 5 Hz case, it shows the uneven output generation from 100 V to 400 V which is resulted from the constructive and destructive interference. However, for the 1-cell at 25 Hz case, it shows relatively even output generation. As the author calculated the average power of them, 5-cell at 5 Hz case generates around 1.4 times higher electrical output than 1-cell at 25 Hz case. To supplement this difference, I think author should add the capacitor charging behavior under these two cases for emphasizing this deviation, because it is one of the important experiments for the novelty.

Response: Thank you very much for your very insightful comments.

To further compare the difference between the 1-cell at 25 Hz and 5-cell at 5 Hz, we have conducted the charging experiments as you suggested. According to our results (**Figure S25** and **Table S2**), both setups achieved ~11% energy storage efficiency with the 200-cell MSC arrays after charging 30 s. However, 5-cell DEG at 5 Hz has a higher output power, so that more energy is stored at the same time, validating its performance superiority. We have added the relevant discussion on **Page 22, Lines 10-20**.

Supplementary Figure 25| Charge-discharge of 200-cell MSC arrays with different DEGs under different operating conditions. a-c Output voltage of the different DEGs (a) one DEG cell at 5 Hz, (b) one DEG cell at 25 Hz, and (c) one DEG panel consisting of 5 DEG cells at 5 Hz. **d** Discharge curves of the 200-cell MSC arrays at a configuration of $10P \times 20S$ under a current of $5 \mu A$, after charging for 30 seconds by the different DEGs.

5) In Figure S4, the maximum average power under the DEG cell, DEG panel, and DEG array is shown. First of all, I think the author needs to make legend in the graph. Right now, it is hard to clarify the experimental data due to absence of the legend. Furthermore, it seems that the optimal resistance, where the load resistance under highest average power, is higher than the previous researches, which shows the hundreds of $k\Omega$ to $30 M\Omega$. What will be the reason of this distinct behavior on it? Finally, it seems that the optimal resistance is decreasing, as the platform is changed from DEG cell to DEG array. What should be the reason of this behavior? The reference which the author can check is written below:
 [1] Jang, Sunmin, et al. "Beyond metallic electrode: spontaneous formation of fluidic electrodes from operational liquid in highly functional droplet- based electricity generator." *Advanced Materials* 36.35 (2024): 2403090.

[2] Zhang, Nan, et al. "A droplet- based electricity generator with ultrahigh instantaneous output and short charging time." *Droplet* 1.1 (2022): 56-64.

[3] Zhang, Nan, et al. "Performance transition in droplet-based electricity generator with optimized top electrode arrangements." *Nano Energy* 106 (2023): 108111.

[4] Wang, Lili, et al. "Monolithic integrated flexible yet robust droplet- based electricity generator." *Advanced Functional Materials* 32.49 (2022): 2206705.

Response: Thank you very much for your very valuable comments.

We are sorry for missing the legend in the graph. As per your suggestions, we have added the appropriate legend to clarify the data presentation.

Supplementary Figure 9 | Average current and power against load resistance for various DEG devices. **a** Equivalent circuit for measuring the output current of a DEG with load resistance R_L through the voltage divider method. In all our tests, $R_S = 1 \text{ M}\Omega$ and $R_{tip} = 100 \text{ M}\Omega$. **b-d** Average current and power of the (b) DEG cell, (c) panel, and (d) panel arrays. The average voltage U_{RMS} is calculated according to Equation (2). The average current is $I_{RMS} = U_{RMS}/R_S$ and average power $P_{RMS} = I_{RMS}^2(R_S + R_L)$.

For an electricity generator/power supply, its output power is maximized when the load resistance is equal to its internal resistance. Several factors influence the internal resistance of a DEG, including triboelectric polymer type and thickness, device structure and water properties. As demonstrated in the references you recommended, when the ionic

concentration of water decreases, the internal resistance of the DEG increases accordingly, and the optimal load resistance also rises to around 30 MΩ. In our case, we also used deionized water, and similarly, the optimal resistance for our **GBE-DEGs** (i.e., the general **global** bottom electrodes as in the literature) was also around 30 MΩ (**Figure S9b**). However, in the case of LBE-DEG (localized bottom electrodes), the optimal resistance increased to be around 100 MΩ, which can be attributed to the reduced bottom electrode area. The reduction in the bottom electrode area increases the resistance between the top and bottom electrodes, which may contribute to a higher internal resistance of the device.

It is worth mentioning that not only our LBE-DEGs have large optimal load resistance exceeding 100 MΩ. A recent paper [Zhang, H. *et al.* Dynamical Mechanism for Reaching Ultrahigh Voltages from a Falling Droplet. *Advanced Functional Materials* **34**, 2315912 (2024).] reports a **GBE-DEG** with optimal resistance reaches as high as 200 MΩ. So, a comprehensive knowledge about the internal resistance of DEGs still needs in-depth exploration in the future.

A DEG panel/array is made up of parallel connection of multiple individual DEG cells. The overall internal resistance of the DEG panel/array is thereby also the “parallel combination” of that of all the contained DEG cells. Therefore, the DEG panel/array has lower internal resistance than the DEG cells. Accordingly, the optimal load resistance also decreases. Notably, the GBE structure does not exhibit the same parallel configuration (due to the already shared large-area global bottom electrode), which explains the minimal variation in optimal resistance observed in **Figure S9b-c**.

6) In Figure S5, it seems that the output is decreased as the droplet continuously impinges for 3 hours. However, in the general knowledge, the electrical output requires to be increased with the surface charging resulted from the continuous liquid-solid contact electrification along the DEG operation, as shown in previous study (Xu, Wanghui, et al. "A droplet-based electricity generator with high instantaneous power density." *Nature* 578.7795 (2020): 392-396.). What would be the reason of this output degradation behavior?

Response: Thank you very much for your very valuable comments.

The increased electrical output is a very common phenomenon in relevant DEG papers. Typically, the output stabilizes after an initial period, and most recent studies focus on the stable performance. For this reason, we also only reported the stable period in our original manuscript. In the revision, we have reported all the periods to indicate output evolution, and extended the stability tests from 3 hours to 8 hours, as shown in **Figure S10**.

Supplementary Figure 10 | Long-term stability test of the DEG device. a, c Output voltage-time curves of the DEG cell driven by (a) DI water and (c) tap water. In each case, after a test time of 8 hours, the DEG cell was cleaned though wiping with paper tissue. b, d Time-resolved average power P_{RMS} and power retention of the DEG cell driven by (b) DI water and (d) tap water.

We think the observed decrease in output power during continuous operation is due to two main factors. First, the accumulation of moisture on the device surface may interfere with droplet sliding. Second, prolonged droplet impact can leave residues on the polymer surfaces. This degradation is more severe for tap water than deionized water, likely due to the higher impurity concentration in tap water. However, we emphasize that this should not be considered permanent damage. The performance can be fully restored through simple cleaning of the polymer surface with paper tissue. For details, please refer to our response to Comment (1) of Reviewer #3. We have included the relevant discussion in the revised manuscript (**Page 10, Line 31 to Page 11, Line 7**).

7) In Figure 4, the author shows the DEG array composed of 6 DEG panel, and it seems that the RMS power is increasing as the number of DEG panels increases. What will happen on CP,C and CP,D, as the size of the DEG panel increases and number of DEG panels increases?

Response: Thank you very much for your very valuable comments.

As the size and the number of DEG panels increase, the $C_{P,D}$ will also increase due to the increased number of cells connected in parallel. However, it is hard to directly quantify

C_{PC} for the DEG panels because of the limitation of our fitting model (Equation S3 for V_{peak} against various capacitance C). The output voltage of a DEG panel is severely degraded and/or strongly irregular (**Figure 3e** and **S15-16**) due to the interference between DEG cells, which makes it difficult to extract the intrinsic V_{peak} . Consequently, it is challenging to obtain from Eq. (S3) the accurate value of C_{PC} for DEG panels. Nevertheless, according to **Figure 2e**, we may speculate that the C_{PC} also increases with the size and number of DEG panels because the overall S_{BE} increases. Despite this, our DEG panel array achieves the best performance reported to date.

8) For the micro-supercapacitor (MSC) array charging demonstrated in Figure 6, how long will it take the MSC to be charged to the level for practical utilization with single DEG array?

Response: Thank you very much for your very valuable comments.

With 30-cell panel arrays, we have demonstrated the practical utilization in our demo videos and have a discussion in the original manuscript, now shown on **Page 24 Lines 16-30**: “With a charging time of 30 seconds, the 200-cell MSC arrays were able to continuously light an LED (Cree, XLamp, MX-6 LEDs) for 35 seconds (in the discharging configuration of $10P \times 20S$, Video S1), while the 400-cell MSC arrays (in the discharging configuration of $10P \times 40S$) extended this duration to 60 seconds (Video S2). Furthermore, after a 30-second charging period, the 200-cell arrays could power a hygrometer (INF, Mini LCD Hygrometer) continuously for approximately 40 seconds (Video S3).”

With a 5-cell DEG panel, the 200-cell MSC array can store more than 230 μJ energy with an output voltage exceeding 1.5 V after charging for 30 seconds (**Figure S25c-d** and **Table S2**), which is sufficient to power small electronic components.

With a single DEG cell, the 200-cell MSC array can store over 50 μJ energy with an output voltage above 0.7 V after charging for 30 seconds (**Figure S25a,d** and **Table S2**). To power small electronic components, a longer charging time, typically more than 60 seconds, is required.

9) Furthermore, it seems that the MSC array with 400 cells possess lower capacitance than the commercial capacitor, which are widely utilized for the energy harvesting researches. What will be the advantageous aspect of using MSC array rather than the commercial capacitor for the aspect of the practical utilization? Also, why the MSC array in this manuscript possesses slow discharging behavior even under the low capacitance? I think the readers who are not familiar with the supercapacitor will have difficulties to understand this

manuscript and experimental data right now. The author must provide kind and detailed explanations on each data for the readers' understanding.

Response: Thank you very much for your very valuable comments.

From a practical utilization perspective, our MSC array exhibits higher energy storage efficiency (ESE) compared to commercial capacitors, which means more energy is collected from the same output energy of the DEGs. As compared with most commercial capacitors, our MSC arrays have a smaller capacitance but a much higher working voltage (up to 640 V). As discussed in the manuscript, both small capacitance and high working voltage contribute to the increase of stored energy within a fixed charging time of DEGs, as discussed in the original manuscript, now shown on **Page 17 Lines 26-30** in revision: “In theory, the stored energy is $E_C = 1/2 CV^2 = 1/2 QV$. Within a certain charging time t that is much longer than the period T of DEG output pulses, the energy harvested by the DEGs is constant, and the charge Q transferred from the DEGs to the capacitors could also be supposed to be constant. As $Q = CV$, small capacitance C could lead to high voltage V , and hence large energy E_C and high ESE.”

With a small capacitance, the MSC array can store more energy *within a fixed charging time*. Because more energy is stored in the charging period and the discharging process uses the same discharging current, a slower discharging behavior is observed. Please be aware in this case, the MSC arrays are *not fully charged*, but just *charged within a fixed time* by DEGs. This is essentially different from the general galvanostatic charging-discharging test of supercapacitors, where the devices are fully charged (with *charging time increasing with capacitance*) so that devices of large capacitance can store more energy and attain longer discharging time.

10) Generally, for the spontaneous operation of the DEG, the target condition should involve rainfall. Considering that raindrops contain various ions and chemical components, they will inevitably affect the conductivity of the droplets, leading to changes in the output generation behavior. In this context, I believe the author should demonstrate the MSC charging behavior using rainwater (or a lab-prepared raindrop-mimicking liquid) and evaluate the electronic device operation under these conditions, as this paper focuses on the practical utilization of the DEG.

Response: Thank you very much for your very valuable comments.

As you correctly pointed out, different droplet compositions can affect the output performance of DEG devices, with raindrops generally causing considerable performance

degradation. We have measured the output performance of the DEG cell and panel arrays with raindrops (thanks to your suggestion, we use the lab-prepared raindrop-mimicking liquid) as shown in **Figures S27 and Table S3**.

Supplementary Figure 27| Test of charging a 200-cell MSC array by a 30-cell DEG panel array driven by DI water and rainwater. a, b Output voltage of the 30-cell DEG panel array driven by (a) DI water and (b) rainwater. **c** Discharge curves of the 200-cell MSC array at a configuration of $10P \times 20S$ under a constant current of $5 \mu A$, after the MSC array has been charged for 30 seconds by the DEG panel array with different types of water.

Despite the decreased performance as compared with DI water, the DEG panel arrays with raindrop-like liquids are still able to continuously power electronic devices. Specifically, after 30 seconds of charging by the 30-cell DEG panel array, the 200-cell MSC arrays stored over $950 \mu J$ of energy with raindrops (compared to $1250 \mu J$ with DI water) and were able to light an LED for around 25 seconds (**Video S4**) and power a calculator for around 70 seconds (**Video S5**). We have added the corresponding discussion to the revised manuscript (**Page 24, Lines 22-30**).

11) It shows typos in the Figure of supplementary information. The author needs to check overall manuscript to avoid typos.

Response: Thank you very much for your very careful comments.

We sincerely apologize for the typos and missing information in the previous version. We have carefully reviewed the entire manuscript and supplementary materials to correct these issues and minimize such mistakes.

Thank you again for your very careful comments to help us improve the quality of this manuscript.